# Local changes in potassium ions regulate input integration in active dendrites

**Malthe S. Nordentoft[1], Naoya Takahashi[2], Mathias S. Heltberg[1], Mogens H. Jensen[1], Rune N. Rasmussen[3]\*, Athanasia Papoutsi[4]\***

**1** Niels Bohr Institute, University of Copenhagen, Copenhagen, Denmark, **2** University of Bordeaux, CNRS, Interdisciplinary Institute for Neuroscience (IINS), UMR 5297, Bordeaux, France, **3** Center for Translational Neuromedicine, University of Copenhagen, Copenhagen, Denmark, **4** Institute of Molecular Biology and Biotechnology, Foundation for Research and Technology—Hellas, Crete, Greece

\* rune.rasmussen90@gmail.com (RNR); papoutsi@imbb.forth.gr (AP)

**Data Availability Statement:** All models used were implemented using Python version 3.10 and NEURON version 8.2. All models are available on

## Abstract

During neuronal activity, the extracellular concentration of potassium ions ($[K^+]_o$) increases substantially above resting levels, yet it remains unclear what role these $[K^+]_o$ changes play in the dendritic integration of synaptic inputs. We here used mathematical formulations and biophysical modeling to explore the role of synaptic activity-dependent $K^+$ changes in dendritic segments of a visual cortex pyramidal neuron, receiving inputs tuned to stimulus orientation. We found that the spatial arrangement of inputs dictates the magnitude of $[K^+]_o$ changes in the dendrites: Dendritic segments receiving similarly tuned inputs can attain substantially higher $[K^+]_o$ increases than segments receiving diversely tuned inputs. These $[K^+]_o$ elevations in turn increase dendritic excitability, leading to more robust and prolonged dendritic spikes. Ultimately, these local effects amplify the gain of neuronal input–output transformations, causing higher orientation-tuned somatic firing rates without compromising orientation selectivity. Our results suggest that local, activity-dependent $[K^+]_o$ changes in dendrites may act as a "volume knob" that determines the impact of synaptic inputs on feature-tuned neuronal firing.

## Introduction

Throughout the nervous system, neuronal activity and ionic changes in the extracellular environment are bidirectionally linked. Yet, extracellular ion changes are not traditionally considered an integral part of neuronal signaling and information processing. Amidst the activity-dependent fluctuations of extracellular ionic concentrations, $K^+$ ions emerge as particularly intriguing due to their pivotal role in shaping neuronal excitability and membrane potential ($V_m$). At rest, the extracellular concentration of $K^+$ ($[K^+]_o$) in the brain is normally between 2.7 and 3.5 mM [1–3]. It has been experimentally shown that during sensory stimulation, motor network activity, sleep oscillations, or behavioral state transitions, $[K^+]_o$ increases by 0.25–2 mM [4–14], while it can rise up to 7–12 mM during hypersynchronous neuronal activity [15–17]. The $[K^+]_o$ increase weakens the outward $K^+$ driving force, resulting in a less negative $K^+$ reversal potential ($E_{K^+}$), which powerfully affects the $V_m$, excitability, and firing

Git: github.com/malthenielsen/potassium_hotspots or Zenodo (DOI: 10.5281/zenodo.14054295).

**Funding:** R.N.R. acknowledges support from the Lundbeck Foundation (R370-2021-764). M.S.H acknowledges support from the Lundbeck Foundation and (R347-2020-2250) and the Novo Nordisk Foundation (NNF23OC0085907). M.H.J. acknowledges support from the Independent Research Fund Denmark (9040-00116B) and the Novo Nordisk Foundation (NNF20OC0064978 and NNF24OC0089788). M.S N. acknowledges support from Novo Nordisk Foundation (NNF20OC0064978). A.P. acknowledges support from the Theodore Papazoglou FORTH Synergy Grants. N.T. acknowledges support from French National Centre for Scientific Research (CNRS), the framework of the University of Bordeaux's IdEx "Investments for the Future" programs (2020 IdEx Junior Chair; GPR BRAIN_2030), Conseil régional Nouvelle-Aquitaine (Bordeaux Neurocampus Junior Chair), the ATIP-Avenir program, Fondation Schlumberger pour l'Education et la Recherche (FSER202401018842), Brain Science Foundation, and Research Foundation for Opto-Science and Technology. None of the sponsors or funders played any role in the study design, data collection and analysis, decision to publish, or preparation of the manuscript.

**Competing interests:** The authors have declared that no competing interests exist.

patterns of neurons [11,14,18–25]. A major source contributing to the [K$^+$]$_o$ changes at the synaptic level is K$^+$ efflux from excitatory glutamatergic receptors, and in particular from NMDA receptors, when compared to other calcium- or voltage-dependent potassium channels [16,21,26]. Importantly, previous work has shown that such postsynaptic NMDA receptor-mediated K$^+$ efflux is highly localized, and can signal to presynaptic axons [21,27], pointing to local [K$^+$]$_o$ changes acting as a modulator of presynaptic transmission. Despite the experimental evidence that synaptic changes can be highly localized, we still lack the experimental tools to systematically disentangle the effect of activity-dependent [K$^+$]$_o$ changes at such a fine-scale level. In addition, whether the activity-mediated [K$^+$]$_o$ changes can modulate postsynaptic integration of synaptic inputs remains elusive.

Pyramidal neurons possess elaborate dendritic trees that contain a variety of voltage-dependent ion channels such as Na$^+$, K$^+$, and Ca$^{2+}$ channels. These channels allow for complex, active responses to synaptic inputs, including the generation of local action potentials, called dendritic spikes [28–31]. The spatial arrangement of synaptic inputs is an important factor for determining dendritic spike initiation [32–38]. These dendritic spikes support the nonlinear summation of synaptic inputs, which in turn alter the neurons' input–output function and can enhance sensory feature selectivity in vivo [39–42]. Interestingly, artificial, pharmacological manipulation of dendritic K$^+$ currents affects dendritic excitability and dendritic spikes, indicating that K$^+$ currents can act as a regulator of dendritic integration [43,44]. Based on the above findings, we here hypothesize that physiological, synaptic activity-mediated changes in [K$^+$]$_o$ can locally regulate the active dendritic properties and thus shape the nonlinear integration of inputs and sensory processing.

To test the hypothesis, we used proof-of-concept mathematical formulations and biophysical modeling to investigate the effect of the reported physiological [K$^+$]$_o$ changes. We focus on the organization of excitatory synaptic inputs tuned to the stimulus orientation on dendritic branches of a pyramidal neuron in the visual cortex [42], as we have previously experimentally shown that visual cortical responses are dynamically regulated by [K$^+$]$_o$ and brain state [14]. Using statistical analysis, as well as abstract and morphologically detailed biophysical models of dendrites and neurons, we show that the arrangement of tuned inputs determines the magnitude of activity-dependent [K$^+$]$_o$ changes in dendrites. Specifically, dendritic segments with similarly tuned synaptic inputs can attain substantially higher [K$^+$]$_o$ elevations than segments with diversely tuned inputs. These [K$^+$]$_o$ elevations in turn depolarize the $E_K{}^+$ which increases the reliability of dendritic spikes and prolongs their duration, but without compromising their stimulus selectivity. Ultimately, these local effects amplify the gain of neuronal input–output transformations, leading to higher firing rates at the soma without affecting feature selectivity. Overall, our results suggest a prominent role for local, activity-dependent, dendritic "[K$^+$]$_o$ hotspots" [21,27] in shaping dendritic integration of synaptic inputs. Importantly, using the visual cortex as a framework of study and grounded by experimental data, we provide a proof-of-concept that grouping of co-tuned inputs creates dendritic [K$^+$]$_o$ hotspots that regulate dendritic integration, a property can be generalized to other sensory features, brain regions, or animal species, with the requirement that synaptic inputs are clustered on a spatial and temporal scale [32,42,45–50].

## Results

### Extracellular space size, intracellular potassium concentration changes, and stimulus orientation shape $E_K{}^+$ shifts

To identify the expected $E_K{}^+$ shifts under different conditions, we first undertook an analytical approach that did not include simulation of the membrane potential dynamics. We

hypothesized that dendritic segments with similarly tuned inputs can attain higher $[K^+]_o$ changes and $E_{K^+}$ shifts than segments with diversely tuned inputs and that this could, in turn, affect dendritic excitability (**Fig 1a**). To investigate this, we sampled the synaptic activity of a hypothetical 10 μm segment, populated with 7 to 13 synapses, yielding a synaptic density comparable to the neocortex [51–53]. We chose this length scale because functional clusters, comprised of inputs with similar tuning preferences, typically are formed within 5 to 10 μm [32,45–47]. To model feature-tuned synaptic inputs, we used visual orientation tuning as our framework. Previous work identified dendritic segments receiving inputs with similar or

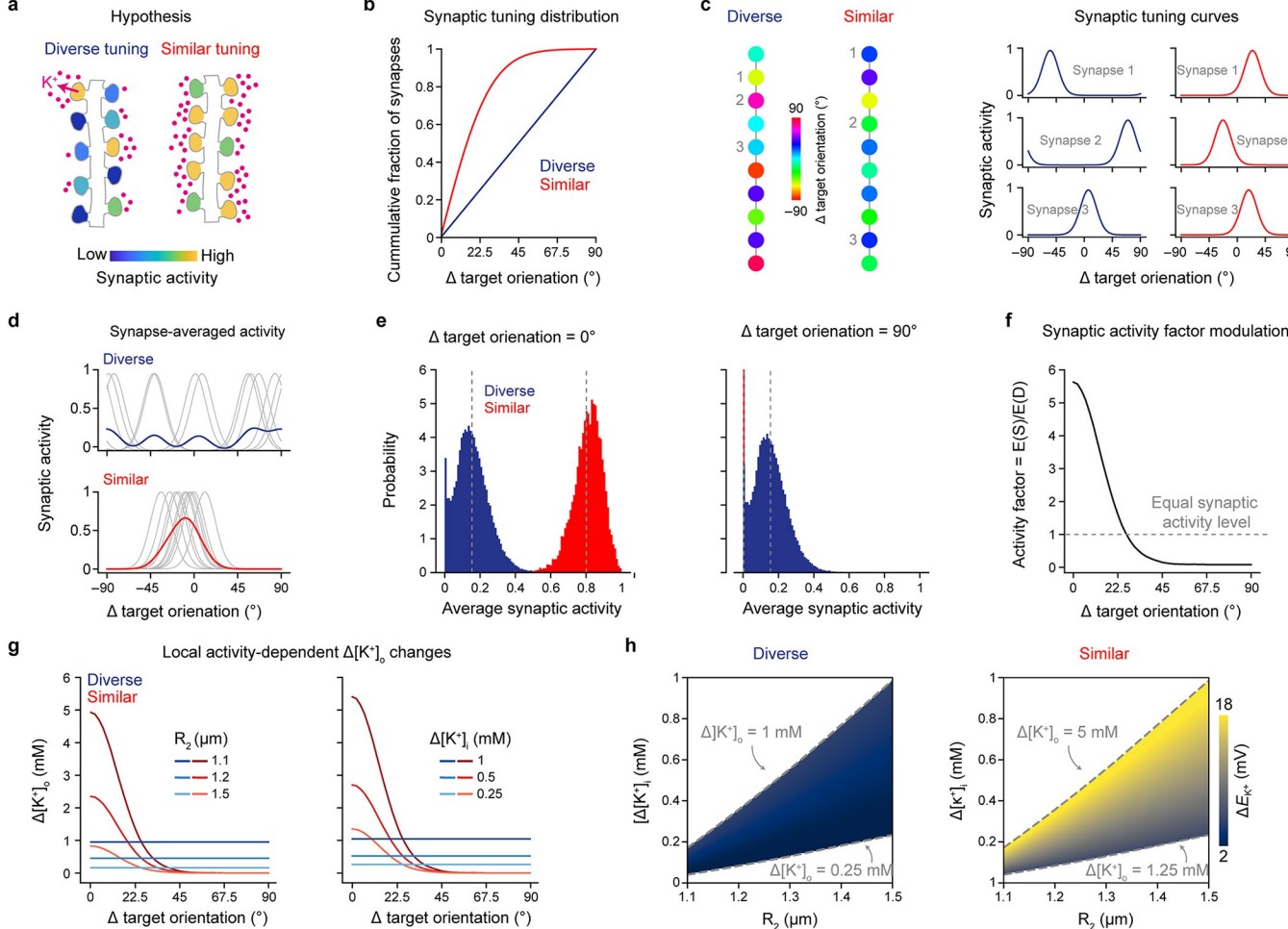

**Fig 1. Dendritic segments with similarly tuned inputs support higher $[K^+]_o$ changes and $E_{K^+}$ shifts.** (a) Diagram of the proposed hypothesis that dendritic segments with similarly tuned inputs can attain higher $[K^+]_o$ increases than segments with diversely tuned inputs. (b) Cumulative fraction of synapse orientation preferences relative to target orientation for dendritic segments with diverse or similar orientation tuning. Reproducing results of [42]. (c) Example dendritic segments with diverse and similar input tuning regimes (left). Synaptic tuning curves for different orientation-tuned stimuli, as per [55], normalized by the maximum activity (right). Note that the x-axis corresponds to stimulus orientation relative to target orientation. The resulting activity shows the tuning curves for the synapses indicated on the corresponding dendritic segment on the left. Based on the distributions of panel b and the synaptic tuning curves, we acquired a statistical approximation of the expected $E_{K^+}$ shifts without explicitly modeling membrane dynamics. (d) Example synapse-averaged tuning curves for dendritic segments from the diverse (top) and similar (bottom) tuning regimes. Tuning curves of individual synapses are in gray and average is in color. (e) Probability distributions of average synaptic activity for the diverse and similar tuning regimes for the target (left) and orthogonal to the target (right) orientations. Dotted lines indicate expectation value. (f) Synaptic activity factor, defined as the ratio between expected synaptic activity for similarly—**E** (S) and diversely—**E** (D) tuned segments as a function of stimulus orientation relative to target orientation. (g) $\Delta[K^+]_o$ as a function of stimulus orientation relative to target orientation for the diverse and similar tuning regimes. Left: different extracellular space radii ($R_2$) with constant $[K^+]_i$ reductions ($\Delta[K^+]_i = 1$ mM). Right: different $\Delta[K^+]_i$ with constant $R_2$ ($\sqrt{2 \mu m}$). (h) Heat maps showing the range of $\Delta E_{K^+}$ for the diverse (left) and similar (right) tuning regimes as a function of $\Delta[K^+]_i$ and $R_2$ at the target orientation. Dotted lines indicate the corresponding $\Delta[K^+]_o$ range.

diverse orientation tuning [42,54]. To capture this, we sampled synaptic orientation preferences from a circular normal distribution for the similarly tuned regime and from a uniform distribution for the diversely tuned regime [42], and assigned tuning curves to individual synapses (**Fig 1a and 1c**). From the sampled segments, we obtained synapse-averaged tuning curves and activity distributions (**Fig 1d and 1e**). Using these, we derived an activity factor for each orientation, signifying the ratio of expected synaptic activity in segments with similar versus diverse tuning preferences (**Fig 1f**; see Methods). This showed that for stimulus orientations close to the target orientation, here arbitrarily chosen as the somatic preferred orientation set at 0˚, (Δtarget orientation: 0–20˚), synaptic activity levels are 2 to 5 times higher in segments with similarly tuned inputs than in segments with diversely tuned inputs. Contrary, for orientations far from the target orientation (Δtarget orientation: 45–90˚), activity ceases almost entirely in segments with similar tuning while it remains constant for segments with diverse tuning, yielding activity factors below 1.

We then asked how these synaptic activity patterns are manifested in $[K^+]_o$ and $E_K^+$ changes. Due to experimental limitations, there is a lack of knowledge about singular ionic flux over the multiple ionic channels that support the in vitro and in vivo documented potassium concentration changes. Thus, we adopted a mean-field approach that allowed us to relate the experimental recordings of extracellular potassium to the model while keeping to the number of assumptions at a minimum. Specifically, in our approach we assumed the following conditions to be true: (1) Local increases in $[K^+]_o$ are proportional to synaptic activity levels; (2) local increases in $[K^+]_o$ are caused by $K^+$ efflux from the intracellular space; and (3) $[K^+]_o$ change is stable in space and time along the short dendritic segment. The latter is motivated by the fact that (a) $K^+$ ions in the extracellular space, of the spatial and temporal scale considered here (approximately 10 μm and approximately 300 ms after synaptic activity onset, respectively), can be described as well-mixed due to the high $K^+$ free diffusion rate ($D_{K^+_{free}} = 1.96 \frac{\mu m^2}{ms}$) [56,57]; and (b) extracellular potassium remains elevated for timescales of several hundreds of ms, as documented in in vitro experiments [58,59] (see Methods for detailed analysis). Following these assumptions, we approximated the $[K^+]_o$ changes of dendritic segments by multiplying the intracellular $K^+$ concentration ($[K^+]_i$) change by the volume fraction between the intra- and extracellular space:

$$\Delta[K^+]_o = |\Delta[K^+]_i| \frac{V_{In}}{V_{Ext}} = |\Delta[K^+]_i| \frac{R_1^2}{R_2^2 - R_1^2} \tag{1}$$

Here, $\Delta[K^+]_o$ and $\Delta[K^+]_i$ are the changes in $[K^+]_o$ and $[K^+]_i$, respectively. The dendritic segment is considered as a cylinder with radius $R_1$, encapsulated by the extracellular space, also described as a cylinder with radius $R_2$ (see Methods, Eq 5). By keeping the dendritic diameter constant ($R_1 = 1$ μm) [60,61], we limit our parameter space to 2 parameters: $\Delta[K^+]_i$ and $R_2$. Although we do not have data on the local dendritic $[K^+]_i$ changes, nor do we know the exact extracellular space size in the vicinity of dendritic segments, we can use Eq 1 to estimate $[K^+]_o$ changes by varying $\Delta[K^+]_i$ and $R_2$ within realistic ranges [62–64]. Note that as $R_2 \to R_1$, $\Delta[K^+]_o$ increases as $\frac{1}{R_2^2 - R_1^2}$ approaching infinity, and becomes undefined for $R_2 = R_1$. As $R_2 \to \infty$, $\Delta[K^+]_o$ goes towards zero asymptotically. By multiplying $\Delta[K^+]_i$ for the diversely tuned input regime by the synaptic activity factor (**Fig 1f**), we also obtain relative estimates of $[K^+]_o$ changes in segments receiving similarly tuned inputs. $\Delta[K^+]_o$ changes are positively correlated with the $\Delta[K^+]_i$ (**Fig 1g, right**), inversely correlated with the size of the extracellular space (**Fig 1g, left**) and, for the similar input regime, higher for orientations close to the target orientation but lower for orientations far from the target (**Fig 1g**).

Focusing on activity for stimuli presented at Δtarget orientation = 0˚, we next identified the $\Delta[K^+]_i$ and $R_2$ parameters that constrained the $\Delta[K^+]_o$ for the diversely tuned inputs to a range of 0.25–1 mM (**Fig 1h, left**), to reflect the changes measured by in vivo $[K^+]_o$ measurements using microelectrodes [5–7,9,10,12–14,18,65], a technique which averages the measured concentration in space and time and likely underestimates true local $[K^+]_o$ changes [66–69]. This set of parameters when applied to the similarly tuned input regime results in $\Delta[K^+]_o$ changes in the range 1.25–5 mM (**Fig 1h, right**). By converting the $\Delta[K^+]$ into shifts in $E_K^+$ via the Nernst equation:

$$\Delta E_{K^+} = -26.7 \, ln \frac{[K^+]_i - \Delta[K^+]_i}{[K^+]_o + \Delta[K^+]_o} - (-)26.7 \ln \frac{[K^+]_i}{[K^+]_o} \tag{2}$$

and using this parameter set, we evaluated the range of the $\Delta E_K^+$ shifts. For diverse inputs, the $\Delta E_K^+$ is within the range of 1.6–5.9 mV above the resting $E_K^+$, and for similarly tuned input, the respective range is 6–18 mV (**Fig 1h**). Based on this analytical approach, in the following sections, we use these ranges in $\Delta E_K^+$ as a proxy for changes in $[K^+]_o$, when we explore their impact on dendritic integration in biophysical models with different levels of abstraction. Together, these data suggest that dendritic segments receiving synaptic inputs with similar tuning preferences can attain substantially higher $[K^+]_o$ and $E_K^+$ changes compared with segments receiving diversely tuned inputs, yet in a stimulus-specific manner.

## Dendritic spike properties and orientation selectivity for different $E_K^+$ shifts

We then investigated the implications of the identified in **Fig 1** $E_K^+$ shifts might have on dendritic synaptic integration. We focused on dendritic segments with similarly tuned inputs as (1) we documented stimulus orientation-dependent and large (above 15 mV) $E_K^+$ shifts in these segments (**Fig 1g** and **1h**); and (2) previous work has shown that spatial clustering of co-active inputs is an important factor for dendritic spike initiation [32–38]. For this analysis, we implemented a biophysically detailed point model to simulate the $V_m$ of a dendritic segment during synaptic input stimulation, referred to as a "point-dendrite" model due to its similarities with a conventional point-neuron model [70] (**Fig 2a**; see Methods). The point-dendrite model included an array of active ion channels found in the dendrites of V1, as well as AMPA and NMDA receptors [71–73] (**S1 Table**). The number of synapses and their orientation tuning were sampled as in **Fig 1** to simulate a dendritic segment receiving similarly tuned inputs. Synaptic inputs were activated by delivering a stimulation train consisting of 3 events in which AMPA and NMDA receptors were activated with a peak current depending on the synapse's orientation tuning and with a Poisson-distributed delay (mean = 80 ms). This protocol induced $V_m$ dynamics that reflected the stimulus orientation in the absence of $E_K^+$ shifts (**Fig 2b**). Importantly, for orientations close to the target orientation, we also observed regenerative activation of NMDA conductances, leading to NMDA-dependent dendritic spikes (see **Fig 3a**), while for the orthogonal orientation the $V_m$ did not change (**Fig 2b**) [42]. To obtain a mechanistic understanding of how $E_K^+$ shifts might shape orientation tuning, we first tested their effect on dendritic spike properties (see Methods) for a stimulus presented at the target orientation, using different $\Delta E_K^+$ values in the predicted range of **Fig 1h**, ([0–18] mV). $E_K^+$ shifts caused a broadening of dendritic spikes (**Figs 2c, 2d** and **S1**); for example, for a 12 mV $E_K^+$ shift the duration of dendritic spikes increased by approximately 80% compared to no $E_K^+$ shift. This effect persisted in the presence of nonspecific inhibitory (GABA$_A$) input (**S2 Fig**). Furthermore, the amount of excitatory synaptic drive needed to transition the dendritic $V_m$ from subthreshold input summation to suprathreshold dendritic spiking decreased substantially with increased $E_K^+$ shifts (**Fig 2e** and **2f**).

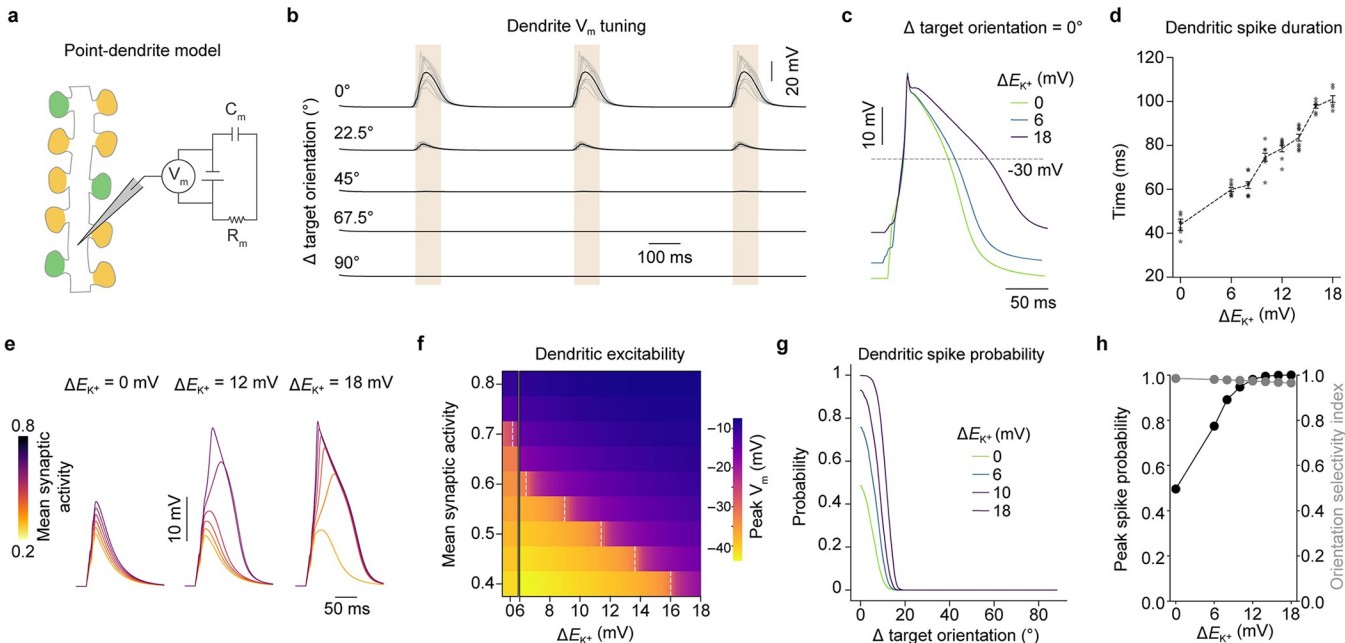

**Fig 2. $E_K^+$ shifts regulate active dendritic properties.** (a) Diagram of the biophysical point-dendrite model. To test the effect of local $[K^+]_o$ elevations, we imposed $E_K^+$ shifts within the interval of 0–18 mV. (b) Example dendrite $V_m$ traces for the similar input tuning regime as a function of stimulus orientation relative to target orientation. Individual trials are in gray and average is in black. Shaded regions indicate synaptic stimulation timing. (c) Example $V_m$ traces highlighting dendritic spike duration (time above –30 mV) as a function of $\Delta E_K^+$ at the target orientation. The $E_K^+$ shift is induced following the first simulation event and is visible in the resting membrane potential of the dendritic segment. (d) Dendritic spike duration as a function of $\Delta E_K^+$ at the target orientation. Error bars are mean ± SEM. ($N = 10$ simulations). (e) Example traces highlighting $V_m$ response as a function of $\Delta E_K^+$ and mean synaptic activity. (f) Heat map showing peak $V_m$ depolarization as a function of $\Delta E_K^+$ and mean synaptic activity. Dotted lines indicate the transition to dendritic spiking. Number of synapses used: $N = 10$. (g) Dendritic spike probability as a function of $\Delta E_K^+$ and stimulus orientation relative to target orientation. Note that $\Delta E_K^+$ values denote shifts at the target orientation and shifts for the rest of the orientations are calculated using the synaptic activity factor. (h) Dendritic spike probability at the target orientation (left axis) and orientation selectivity index (right axis) as a function of $\Delta E_K^+$. See also **S3** and **S4 Figs** and **S1 Table**.

In our model, 3 main parameters control dendritic spike generation: the number of dendritic synapses (N), the average synaptic activity (w), and the $\Delta E_K^+$ (see Methods for details). By simulating different parameter configurations with dendritic spike occurrence as binary output measure, we fitted a function that describes the minimum $\Delta E_K^+$ needed to trigger dendritic spiking given N and w (**S3 and S4** Figs):

$$\Delta E_{K^+}(N, w) = aN + \beta w + v \tag{3}$$

Using Eq (3), we estimated the probability of generating a dendritic spike for a given stimulus orientation (**Fig 2g**). The probability of eliciting dendritic spikes rose drastically as a function of increasing $E_K^+$ shifts (**Fig 2g** and **2f**); for example, the probability of spiking to the target orientation increased from 48% to 97% when shifting the $E_K^+$ by 12 mV. This gain in the dendritic output was similar to the one expected from increasing the number of synapses (**S5 Fig**), with the main difference being that $E_K^+$ modulation is transient, selectively boosting the activity of repetitively active dendrites. Interestingly, the dendritic spiking orientation selectivity was largely constant even though the probability of spiking increased also for orientations away from the target orientation (**Fig 2g** and **2h**; Orientation selectivity index: 0.98, 0.98, and 0.96 for $\Delta E_K^+ = 0$, 6, and 18 mV, respectively). Altogether, these results show that local K⁺ changes in dendritic segments with similarly tuned synaptic inputs prolong dendritic spikes and boost the probability of generating dendritic spikes without affecting their feature selectivity.

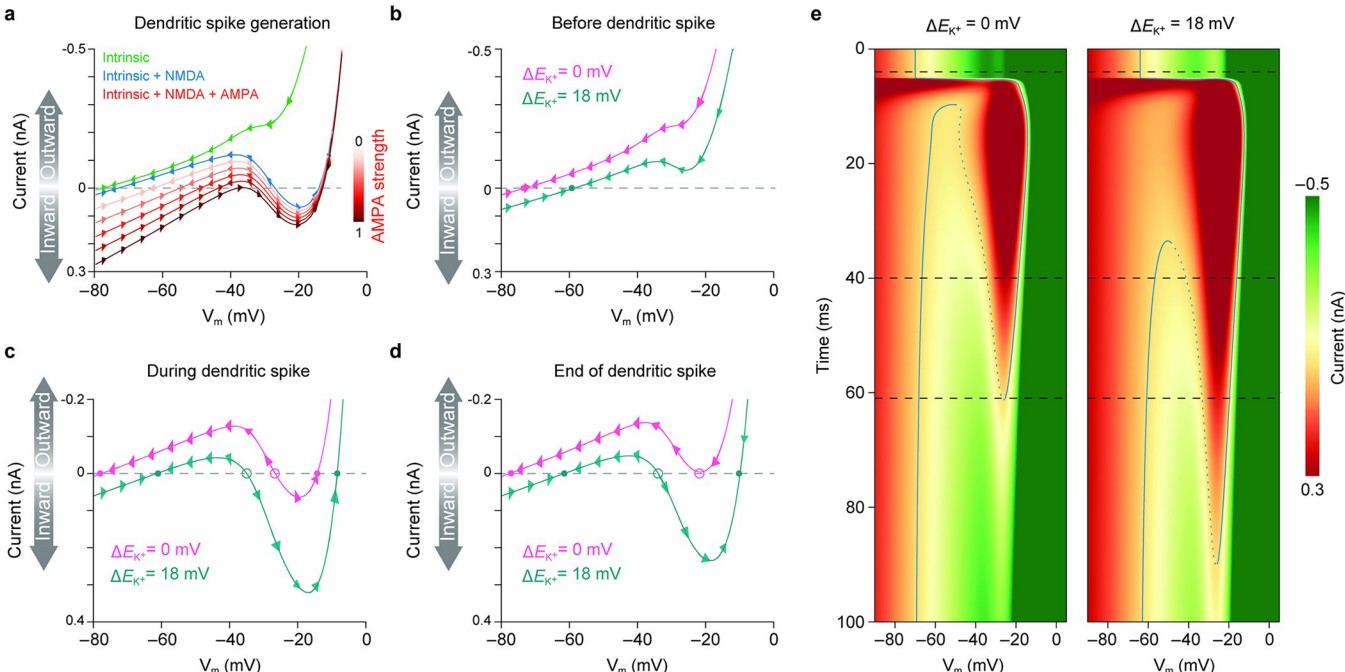

**Fig 3. Current-voltage (I-V) relations and bistability with $E_{K}^{+}$ shift.** (a) I-V curves during the generation of dendritic NMDA spike. Green: down-stable state with only intrinsic ion channels active. Blue: bistable state with intrinsic ion channels and NMDA receptor active. Red: Bi- and up-stable state with intrinsic ion channels and NMDA and AMPA receptors active. (b) Down-stable state before dendritic spike generation without and with $E_{K}^{+}$ shift. Solid points indicate stable fixed points. (c, d) Bistable states during and at the end of dendritic spike without and with $E_{K}^{+}$ shift. (c) Shows the hypothetical system with peak NMDA receptor conductance, without AMPA receptor activation, right before spike initiation, and (d) shows the system when outward and inward currents match for the system without $E_{K}^{+}$ shift around the end of the spike. Solid and open points indicate stable and unstable fixed points, respectively. (e) Heat maps showing the temporal evolution of the I-V landscape without and with $E_{K}^{+}$ shift. The corresponding I-V curves shown in (b–d) are indicated with black dotted lines, and full and dotted blue lines indicate stable and unstable fixed points, respectively. (a–c) Arrows indicate system flow direction. Note that positive inward current depolarizes the membrane, resulting in I–V plot similar to previous work (Major and colleagues [29]). See also **S1 Video**.

## $E_{K}^{+}$ shift alters the current-voltage attractor landscape in dendrites

To understand the general biophysical principles governing the regulation of dendritic spikes by local K⁺ changes, we turned to dynamical systems theory [74]. Dendritic spiking can be described by the instantaneous phase plot of the $V_m$ and $\frac{dV_m}{dt}$. Given that the membrane capacitance is constant, we transformed this to the I-V curve using $\frac{dV_m}{dt} = -\frac{I}{C}$ [29]. With only the intrinsic ion channels active, $V_m$ is attracted to a single down-stable hyperpolarized fixed point corresponding to the resting potential (**Fig 3a**). Inclusion of NMDA receptors changes the system to bistable, and an attractor basin together with a depolarized fixed point, corresponding to dendritic spiking, is created. The voltage barrier preventing the system from transitioning to an up-stable state, where $V_m$ is only attracted to the depolarized fixed point, is overcome by the activation of AMPA receptors. As the activity of AMPA and NMDA receptors wears off, in conjunction with the high activity of K⁺ channels, the system is again pushed back to the down-stable state.

Shifting the $E_{K}^{+}$ alters the I-V curve attractor landscape in 3 fundamental ways (**Fig 3b–3e** and **S1 Video**). First, for the down-stable state, it moves the fixed point to a more depolarized $V_m$, as well as reducing the net outward current across all $V_m$ levels (**Fig 3b**). Second, for the bistable state, it lowers the voltage barrier needed to transition the system to the up-stable state and deepens the attractor basin (**Fig 3c and 3d**). Finally, by reducing the net outward currents at depolarized $V_m$ levels, it prolongs the duration for which the system is in up- and bistable

states (**Fig 3c–3e**). These dynamical system properties are all explained by the weakened K$^+$ driving force through K$^+$ channels and fully predict the $\Delta E_K{}^+$-mediated effects we observed on dendritic spiking (**Fig 2**). Collectively, this analysis demonstrates that the effects of local K$^+$ changes on synaptic integration can be predicted by the altered dynamical system properties of dendritic spikes.

## Local dendritic $E_K{}^+$ shifts induce neuronal firing gain modulation

Our data suggests that the local [K$^+$]$_o$ increases in dendritic segments receiving similarly tuned inputs can boost the reliability and duration of dendritic spikes without compromising feature selectivity (**Fig 2g** and **2h**). Thus, we next sought to investigate the functional consequences this result might have on the somatic neuronal output. For this, we constructed an abstract neuron model using a fractal tree, mimicking the generic compartmentalization and fractal dimensionality of the apical dendritic tree of pyramidal neurons [75,76] and we stimulated distal dendrites with similarly tuned input [38,77,78] (**Fig 4a**). We simulated 3 conditions of $E_K{}^+$ shifts in the stimulated dendrites: no change, small shift, or large shift, based on the [K$^+$]$_o$ changes determined in **Fig 1** (**Fig 4a** and **4b**; see Methods). These 3 levels correspond to $\Delta E_K{}^+$ = 0 mV, $\Delta E_K{}^+$ = 6 mV, and $\Delta E_K{}^+$ = 18 mV, respectively, to represent the case of no contribution as well as the minimum and maximum level used in (**Fig 1**), based on in vivo recordings. In S6 Fig, we also included a similar test using a reconstructed neuron morphology based on a previously published model.

These simulations revealed that somatic firing rates were significantly amplified when $E_K{}^+$ values were shifted locally (**Fig 4c** and **4d**; $P$ = 0.007 and P = 0.003 for small and large $E_K{}^+$ shifts, respectively, one-tailed Student's $t$ test, $N$ = 15 simulations). We determined that this form of gain modulation was consistent with a multiplicative transformation (**S7 Fig**; gain coefficient = 1.64 and 2.73 for small and large $E_K{}^+$ shifts, respectively), similarly to previously shown in the visual cortex [79]. This amplification was similar to the one expected by increasing the number of synapses (**S8 Fig**) with the main difference being that $E_K{}^+$ modulation closely follows the dendritic activity levels, operating in time scales of seconds. Importantly, and in congruence with what we observed in the dendrites (**Fig 2h**), the somatic firing selectivity was not changed by $E_K{}^+$ shifts despite increases in firing rate also for orientations away from the target orientation (**Fig 4c** and **4d**; orientation selectivity index: 0.99, 0.98, and 0.98 for no, small, and large $E_K{}^+$ shifts, respectively; $N$ = 15 simulations). To elucidate how dendritic $E_K{}^+$ shifts cause dendritic gain modulation, we measured the area under the curve of the integrated V$_m$ signal arriving at the base of the dendritic trunk during a stimulation event. To exclude back-propagating action potentials (bAPs), for this analysis, we turned off the voltage-gated sodium channels in the soma and trunk. This showed that the strength of the integrated dendritic signal, received by the soma, rose substantially with increasing $E_K{}^+$ shifts (**Fig 4e**); for example, the signal strength at the target orientation increased by around 12% when imposing large $E_K{}^+$ shifts compared to no $E_K{}^+$ shifts.

Finally, we asked what functional benefits there might be to this K$^+$-mediated gain modulation. A major task of dendrites is to transmit incoming inputs to the soma for output generation. This is an inherent challenge, especially for distal inputs to large neurons, as the voltage signal tends to attenuate as a function of distance traveled and branching points [80,81]. We, therefore, speculated that one function of K$^+$-mediated gain modulation might be to promote neurons transmitting signals over larger dendritic distances. To test this, we manipulated the synaptic input distance to the soma by scaling the entire neuron while measuring the integrated V$_m$ signal at the base of the neuron (**Fig 4f and 4g**). Interestingly, due to the local gain of the synaptic input, the $E_K{}^+$-shifted regime is attenuated less with distance. This, in turn,

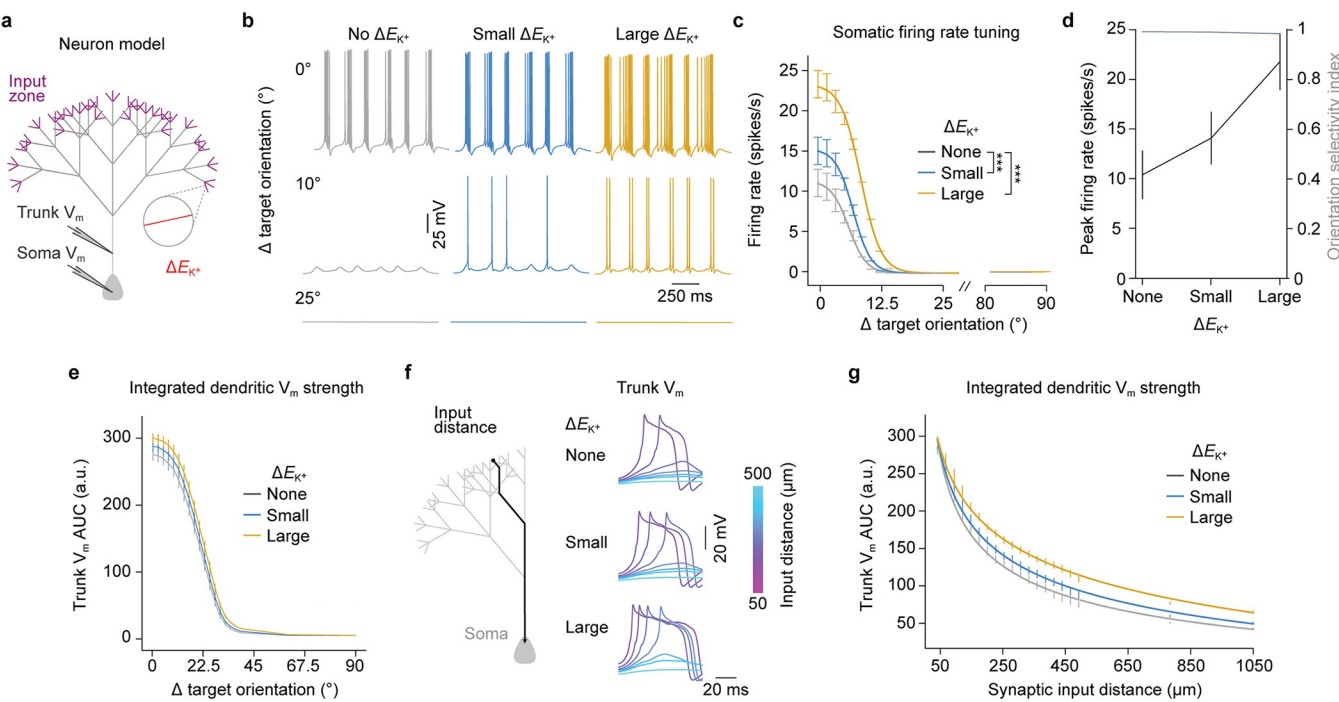

**Fig 4. Local dendritic $E_K^+$ shifts induces neuronal firing gain modulation.** (a) Diagram of the abstract neuron model. The neuron received orientation-tuned synaptic input at the distal dendrites and local $E_K^+$ shifts. (b) Example soma $V_m$ traces obtained as a function of dendritic $E_K^+$ shift magnitude and stimulus orientation relative to target orientation. (c) Soma firing rate tuning curve for different dendritic $E_K^+$ shift magnitudes. Error bars are mean ± SEM. ($N$ = 15 simulations). ***$P = 10^{-16}$ and $10^{-30}$ for small and large $E_K^+$ shifts, respectively, one-tailed Student's $t$ test. (d) Soma peak firing rate at target orientation and orientation selectivity index as a function of dendritic $E_K^+$ shift magnitude. Error bars are mean ± SEM. ($N$ = 15 simulations). (e) Area under the curve for the dendritic trunk $V_m$ measurements as a function of stimulus orientation relative to target orientation and dendritic $E_K^+$ shift magnitude. Error bars are mean ± SEM. ($N$ = 15 simulations). Trunk length was kept constant at 500 μm. Note that bAPs were here abolished by silencing voltage-gated sodium channels in the trunk and soma. (f) Example dendritic trunk $V_m$ traces obtained at the target orientation as a function of synaptic input distance to soma and dendritic $E_K^+$ shift magnitude. (g) Area under the curve for the dendritic trunk $V_m$ measurements at the target orientation as a function of synaptic input distance to soma and $E_K^+$ shift magnitude. Error bars are mean ± SEM. ($N$ = 15 simulations) and solid lines represent fit to the function $f(x) = x^a + c$. ***$P = 0.61$, 0.41, and 0.38 for different $E_K^+$ shifts. See also **S7 Fig** and **S2 Table**.

makes it possible for signals to travel substantially longer, before reaching a similar integrated $V_m$ signal. Based on the power-law fits using the $\chi^2$ fitting method for each $\Delta E_K^+$ condition, we observe that as the neuron grows, the signal in the absence of $E_K^+$ shift is attenuated by a factor $\sim \sqrt[4]{L}$ when compared to the large $E_K^+$ shifts (**Fig 4g**). Together, these results show that local activity-dependent $[K^+]_o$ increases and $E_K^+$ shifts in dendrites enhance the effectiveness of distal synaptic inputs to cause feature-tuned firing of neurons, without comprising feature selectivity.

## Discussion

We have developed mathematical formulations and biophysical models to address the open question of how local activity-dependent changes in $[K^+]_o$ affect dendritic integration of sensory-tuned synaptic inputs. Our work provides three major insights into this question. First, assuming activity-dependent $[K^+]_o$ changes in dendritic segments, the fine-scale arrangement of orientation-tuned synaptic inputs determines the magnitude of these changes; that is, segments with similarly tuned inputs can attain substantially higher $[K^+]_o$ increases than segments with diverse inputs. Second, these $[K^+]_o$ elevations in turn depolarize the $E_K^+$ which enhances dendritic excitability, increasing both the reliability and the duration of sensory-evoked dendritic spikes. Finally, these local dendritic effects promote gain amplification of neuronal input–output functions, resulting in increased somatic responsiveness without

affecting the feature selectivity of the neuron. Our results, therefore, suggest a prominent and previously overlooked role for local activity-dependent changes in K$^+$ concentration in regulating dendritic computations, shedding new light on the mechanisms underlying sensory integration in neurons.

Dendritic processing of synaptic inputs depends on the spatial and temporal organization of the inputs: spatially dispersed and asynchronous inputs are summed linearly, while inputs that are spatially localized and synchronous are summed nonlinearly and can facilitate the generation of dendritic spikes [32–38,82,83]. Here, we propose the novel idea that another important function of grouping co-tuned synaptic inputs close in space is to generate higher activity-dependent [K$^+$]$_o$ increases, creating local and feature-tuned dendritic "[K$^+$]$_o$ hotspots." Such [K$^+$]$_o$ hotspots, potentially attaining up to 5 mM K$^+$ increases relative to baseline (**Fig 1**), can markedly affect dendritic processing by dampening K$^+$ currents. For example, we show here that they can reduce the amount of excitatory drive needed to trigger dendritic spiking, as well as broaden the dendritic spikes (**Figs 2** and **3**). Our results on dendritic integration are supported by previous work showing that K$^+$ channels, highly expressed in dendrites [84], can modulate dendritic properties, albeit using artificial pharmacological or genetic disruption of dendritic K$^+$ channels [43,44,84,85], or global [K$^+$]$_o$ elevations [82]. Here, we introduce the novel concept of activity-dependent, local [K$^+$]$_o$ changes that increase the excitability of dendrites and prolong dendritic spikes.

Regarding the functional role of the [K$^+$]$_o$-dependent dendritic integration modulation, we showed that [K$^+$]$_o$ increases can act as a volume knob and cause multiplicative gain modulation of the neuronal input–output function by boosting the effectiveness of orientation-tuned synaptic inputs (**Fig 4**). This makes the neuron more responsive and could contribute to increasing the signal-to-noise ratio of visual cortical neurons while, importantly, maintaining their orientation selectivity. Compared to long-term plasticity mechanisms, [K$^+$]$_o$-dependent dendritic integration modulation requires minimum usage of resources and is transient, operating in time scales of seconds, selectively boosting the activity of repetitively active dendrites; as such, [K$^+$]$_o$ changes closely follow the overall dendritic activity levels. Indeed, we have previously shown that [K$^+$]$_o$ changes are state dependent, e.g., on whether the animal is in quiescence or locomoting [14], indicating that state-dependent modulation of sensory processing may be supported by [K$^+$]$_o$ changes following the overall activity levels. In addition, the broadening of dendritic spikes could potentially also enhance the capacity of the neuron to integrate temporally delayed excitatory inputs [86] and act as a cellular mechanism involved in short-term memory. Notably, the [K$^+$]$_o$—dependent increase in the reliability of dendritic spikes is of potential high functional importance as dendritic spikes are critical for in vivo learning and behavior [87]. For example, the dendritic response to inputs arriving on the apical dendrites of cortical neurons has been implicated in contextual modulation of sensory processing and perceptual sensitivity [88–90]. A recent report suggests that densely localized thalamic inputs to apical dendrites might regulate such modulation by facilitating dendritic spikes [91]. It is thus tempting to speculate that local activity-dependent [K$^+$]$_o$ increases of spatially clustered similarly tuned inputs to apical dendrites effectively modulate cortical sensory processing, by means of promoting dendritic spikes. In this context, it is important to note that while we chose here to use visual orientation tuning as our framework of study, our proposed mechanistic concept is agnostic to sensory features, brain regions, or animal species. The fundamental requirement for the emergence of dendritic [K$^+$]$_o$ hotspots is that synaptic inputs are activated on a spatially and temporally synchronized scale. Hence, this mechanism could play a role in dendritic computations in diverse brain regions such as the hippocampus, motor cortex, visual cortex, and somatosensory cortex where spatiotemporally synchronized inputs have been observed [32,42,45–50].

Importantly, our prediction on the existence of the local activity-dependent $[K^+]_o$ hotspots can be potentially experimentally tested. The gold standard for measuring $[K^+]_o$ dynamics in the brain is with $K^+$-selective microelectrodes [5,7–10,12–14,65]. However, this technique creates a dead space surrounding the electrode, and can only measure from a single point in space, hence providing poor spatial resolution. Instead, to test our hypothesis, an optical approach like two-photon microscopy with fine-scale synaptic resolution seems ideal. Genetically encoded green fluorescent protein-based $[K^+]_o$ sensors exist [92], which could be combined with a red-shifted genetically encoded voltage indicator [93] to simultaneously monitor $[K^+]_o$ dynamics and synaptic synapse $V_m$ tuning in vivo. We hope that experimentalists in the future will use these advanced techniques to probe the existence of local dendritic $[K^+]_o$ changes, as well as their functions and the cellular mechanisms regulating them, such as astrocyte-mediated $K^+$ uptake [94].

Our approach also comes with its limitations: First, while we focused only on the postsynaptic impact of local $[K^+]_o$ changes, it is reasonable to predict that activity-dependent $[K^+]_o$ hotspots could also affect the presynaptic terminals [18]. Increases in $[K^+]_o$ depolarize axons, which can broaden action potentials and increase presynaptic calcium entry [95,96], leading to enhanced glutamate release and stronger synaptic transmission. Such amplification of local synaptic integration, through pre- and post-synaptic mechanisms, could play important role in neuronal circuit development and long-term potentiation by supporting spike timing-dependent plasticity [97,98]. Second, $[K^+]_o$ changes may modulate the gating properties of specific types of $K^+$ channels, such as the inward-rectifier $K^+$ (Kir) [99] or the slow delayed rectifier (KCNQ1) [100,101], potentially providing another venue for $[K^+]_o$ self-regulating mechanism of dendritic excitability. Third, synapses surrounding the dendritic segments may also participate or be affected by the local $[K^+]_o$ changes: activity of the surrounding synapses, belonging to dendritic branches of the same or different neurons may also contribute to the local $[K^+]_o$. This co-activation of nearby dendrites could either increase the noise levels or locally promote the synergy between synapses, dendrites, and neurons. Fourth, in this work, we did not assess in detail GABAergic inhibition, known to play important roles in regulating dendritic activity [102,103]. Future work could address if local $[K^+]_o$ increases near dendritic segments with similarly tuned excitatory inputs could cause disinhibition by reducing the inward driving force for chloride ions thus increasing neuronal feature selectivity or creating a temporal window for dendritic plasticity. Fifth, the width of the extracellular space can be highly heterogeneous, ranging from 0.04 to 0.5 μm [63,104], yet smaller than the pyramidal neuron dendritic diameter [105,106]. Importantly, the extracellular space itself is a dynamic compartment with timescales ranging from seconds to hours; for example, the extracellular space decreases in response to epileptiform activity [107] and these structural changes can be highly local [63], making the extracellular space size another mean for local modulation of dendritic excitability. Finally, our mean field approximation quantifies average $[K^+]_o$ changes after synaptic activation and thus lacks temporal resolution. We chose this approach because several important parameters have yet to be experimentally determined, including the voltage-dependent fraction of $K^+$ currents of total NMDA currents and the precise geometry of the extracellular space surrounding functionally mapped dendritic segments, which prevents precise simulations of time-dependent $[K^+]_o$ dynamics. We assumed here that the system goes asymptotically towards the well-mixed state following synaptic activation, and we estimate that this is reached approximately 150–170 ms after input onset, but we cannot infer the nature of the spatiotemporal $[K^+]_o$ dynamics before this time point. In the future, experimental knowledge about these parameters would enable us to make a full-scale electrodiffussive model of the extracellular space, improving the temporal resolution of extracellular ion dynamics.

In conclusion, our study is a first step towards unraveling the role of extracellular K$^+$ changes in dendritic integration. Future theoretical and experimental studies are needed to obtain a comprehensive understanding of how local activity-dependent ionic shifts contribute to information processing and computations in dendrites.

## Methods

### Sampling of orientation-tuned synapses of dendritic segments

We defined a dendritic segment as a cylinder with length L = 10 μm and radius $R_1$ = 1 μm [32,45–47]. Each segment was randomly populated with 7 to 13 synapses, yielding synaptic densities comparable to that of neocortical neurons [51–53]. The orientation preference of individual synapses of a dendritic segment depended on the synaptic organization type to which the segment belonged, that is, the diverse or similar input tuning regime. Synapses in segments with similar tuning were assigned an orientation preference using a half-circular normal distribution with σ = 15˚ [42,72,108] and a mean target orientation arbitrarily set to 0˚, while synapses in segments with diverse tuning were assigned orientation preferences using the uniform distribution. Importantly, for both types of input regimes, we could reproduce the synaptic orientation preference cumulative distributions measured previously in the visual cortex of ferrets [42]. Moreover, the orientation preferences of our diverse tuning regime resembled those previously recorded in the visual cortex of mice [45,54]. All synapses, irrespective of the spatial organization regime, had a tuning curve width given by a half-circular normal distribution with $\sigma_{tuning}$ = 11˚ [42,55], and the activity of each synapse spanned from 0 to 1 (**Fig 1c**). This sampling method for the number and tuning properties of excitatory synapses of dendritic segments was implemented throughout this work.

### Synaptic activity factor of dendritic segments

We modeled the activation of synapses of dendritic segments by assuming that all synapses were activated by visual stimulation according to their tuning curve. To assess the expected synaptic activity level of a segment for a given stimulus orientation, we calculated the average activity of all synapses in that segment, w(θ), that depends on the orientation tuning of each synapse and the distribution of orientation preferences in the dendritic segment (**Fig 1d and 1e**). We then obtained the synaptic activity factor by comparing the expectation values from the synaptic activity distributions for the 2 types of dendritic segments. We sampled $10^4$ dendritic segments and used a Gaussian kernel density estimator to determine the probability density function p(x) for the expected synaptic activity distributions for each segment type. The expectation value was calculated as $E(X) = \sum_i x_i p_i$ where x was a linearly spaced set of values between 0 and 1. To investigate the stimulus orientation-dependent modulation of the synaptic activity factor (**Fig 1f**), we repeated this procedure for all stimulus orientations in the interval θ ∈ [0: 90˚].

### Extracellular space surrounding the dendrite segment

The exact size and shape of the extracellular space surrounding functionally mapped, short dendritic segments is currently unknown, so we chose to model it as a cylinder that encapsulates the dendritic segment cylinder. The space in between these 2 cylinders thus constitutes the extracellular space in our investigations, and its volume ($V_{ext}$) is given by:

$$V_{ext} = \pi L (R_2^2 - R_1^2) \tag{4}$$

With $R_1$ and $R_2$ being the radius of the inner and outer cylinder, respectively, both with length L = 10 μm. This can be rewritten to express the volume ratio ($V_R$) as the ratio between the 2 cylinders:

$$V_R = \frac{V_{int}}{V_{ext}} = \frac{R_1^2}{R_2^2 - R_1^2} \qquad (5)$$

For simplicity, synapses were considered as points on the surface of the inner cylinder, and hence, the volume of a synapse was considered part of the dendritic segment volume.

## K⁺ diffusion state

During synaptic excitatory transmission, K⁺ ions are released into the extracellular space primarily through NMDA receptors [16,21,26,27] and move around due to diffusion, although their movement is hindered by various cellular elements within the extracellular space. To account for this hindrance, we define effective diffusion as a function of tortuosity [66]:

$$D^* = \frac{D_{K^+_{free}}}{\lambda^2} = \frac{1.96 \frac{\mu m^2}{ms}}{1.6^2} = 0.76 \frac{\mu m^2}{ms} \qquad (6)$$

Here, $D_{K^+_{free}}$ is the diffusion of free $K^+$ ions in physiological saline, $\lambda$ is a non-dimensional measure of how hindered an ion is in its movement in a given space, and $D^*$ is the resulting diffusion rate. The parameter for $D_{K^+_{free}}$ and $\lambda$ were based on previous work [57,66,109,110].

Here, we assume that diffusion in the radial direction is negligible, as the extracellular space is very small compared to the dendritic segment length. Moreover, we assume closed boundary conditions on the outer cylinder wall, since the extracellular space mainly contacts cellular membranes of neighboring neurons and/or glial cells. Finally, assuming the extracellular K⁺ behaves like a brownian particle, the spatial diffusion of the K⁺ will extent as a multivariate normal distribution with standard deviation $\sigma = \sqrt{2D^*\tau}$ [111,112]. This equates the average distance traveled for an individual particle, after a certain characteristic time $\tau$. Using this equation, we can likewise compute the time it will take for a particle to diffuse at a distance L:

$$L = \sqrt{2D^*\tau} \rightarrow \tau = \frac{L^2}{2D^*} \qquad (7)$$

After this characteristic time $\tau$, we assume that the $[K^+]_o$ is well-mixed and uniform throughout the extracellular space. For the dendritic segment with length L = 10 μm, the characteristic time becomes $\tau \approx 66$ ms. Considering that the majority of K⁺ efflux occurs within $\approx 100$ ms from initial synaptic activation, we predict a peak $[K^+]_o$ elevation along the dendritic segment after ~150–170 ms, for a $\tau \approx 66$ ms. Following, pumps and channels return the K⁺ levels to baseline, yet operating on much lower timescales as seen in in vitro experiments [58,59] (see **S2 Text** and **S9**–**S13 and S15 Figs** for simulations that describe the spatial and temporal properties of K⁺ diffusion in the presence of Na⁺/K⁺ pumps). Thus, we analyzed our models with fixed $[K^+]_o$ changes, with values chosen as described in the section below. This time scale separation, due to the different time constants of the described phenomena, has been successfully applied previously on similar questions [113]. Finally, we assume that the boundary between the dendritic segments is closed, implying a finite ion concentration within the segment. This limits our approach since we do not investigate the effect of spillover K⁺ ions moving along the dendrite, potentially affecting neighboring segments. However, given the average distance traveled for a K⁺ ion between the stimuli considered here (based on Eq 7, the expected movement is $\approx 30$ μm during a period of 600 ms),

only the directly neighboring segments could potentially be affected, thus keeping the ionic shift highly localized (**S10 Fig**).

## Activity-dependent K$^+$ change across the neuronal membrane

To estimate synaptic activity-dependent [K$^+$]$_o$ changes near dendritic segments, we assumed that local increases in [K$^+$]$_o$ are linearly correlated with the expected synapse activity, as a result of K$^+$ efflux from the intracellular space. By scaling the K$^+$ change in the diversely tuned input regime with the stimulus orientation-dependent synaptic activity factor, we obtained an estimate of the K$^+$ change in the similarly tuned regime. Using this, in conjunction with Eq 5, we derived equations for the 2 types of input regimes:

$$\Delta[Diverse\ K^+]_o(\theta, \Delta[K^+]_i) = |\Delta[K^+]_i| V_R \tag{8}$$

$$\Delta[Similar\ K^+]_o(\theta, \Delta[K^+]_i) = \mathbf{E}(\theta)\Delta[Diverse\ K^+]_o(\theta, \Delta[K^+]_i) \tag{9}$$

With $\Delta[K^+]_o$ and $\Delta[K^+]_i$ being the change in extracellular or intracellular [K$^+$], respectively, V$_R$ the volume ratio as defined in 5 and $\mathbf{E}(\theta)$ being the stimulus orientation-dependent synaptic activity factor. Using the Nernst equation, we then can express the shift in $E_K^+$ (see Eq 2).

## Point-dendrite model

We simulated the dendritic segment V$_m$ as an isolated resistance-capacitance circuit, similar to the common practice for point-neuron models [70]. The major differences between point-neuron models and the point-dendrite model developed here stem from the specific ion channel setup and cellular resistance, chosen to mimic dendritic properties [73] (**S1 Table**). The circuit is given by the ordinary differential equation:

$$\frac{dV_m}{dt}C_m = I_{ext} - AI_{int} \tag{10}$$

$C_m$ Is the membrane capacitance, set to $2\ \frac{\mu F}{cm^2}$ to simulate the contribution from the spines [114], $I_{ext}$ is the current generated by synaptic receptors (AMPA and NMDA receptors), and $I_{int}$ is the current generated by intrinsic ion channels (K$_{leak}$, Na$_V$, K$_V$, K$_M$, K$_A$, K$_{Ca}$, C$_{aV}$, and HCN). The latter is multiplied by the surface area, $A$, as these channels are scattered uniformly along the surface of the dendrite. All relevant channel dynamics and specific parameters are included in the S1 Text. The system was simulated using an exponential Euler scheme (cnexp):

$$V_m(t + dt) = V_\infty(t) + (V_m(t) - V_\infty(t))exp\left(\frac{-dt}{\tau_{V_m}(t)}\right) \tag{11}$$

Here, $V_\infty$ is the solution to the equation $\frac{dV_m}{dt} = 0$:

$$V_\infty(t) = \frac{\bar{g}_L E_L + \bar{g}_{Nav}m^3hE_{Na} + \bar{g}_{Kv}nE_{K^+} + \bar{g}_{KA}nlE_{K^+} + \bar{g}_{KCa}nE_{K^+} + \bar{g}_{Cav}m^2hE_{Ca^{2+}} + \bar{g}_{HCN}mE_{HCN} + \sum g_{syn}E_{syn}}{\bar{g}_L + \bar{g}_{Nav}m^3h + \bar{g}_{Kv}n + \bar{g}_{KA}nl + \bar{g}_{KCa}n + \bar{g}_{Cav}m^2h + \bar{g}_{HCN}m + \sum g_{syn}} \tag{12}$$

Where $\bar{g}$ denotes a channel's average conductance, $n^x$ is from the individual channel dynamics controlling the gating and channel-specific conductance, $E_{ion}$ denotes the relevant reversal potential and $\sum g_{syn}E_{syn}, for\ syn \in AMPA, NMDA, GABA_A$ sums the current from the different types of synaptic receptors, were $g_{syn} = -g_{syn}(B - A)f, with\ f = Mg_{block}(V_m)if\ syn ==$ $NMDA, and\ f = 1\ otherwise$. For complete synaptic equations for NMPA, NMDA, and

GABA$_A$ receptors, see **S1 Text**. The membrane time constant is given as follows:

$$\tau_{V_m} = \frac{C_m}{g_m} \tag{13}$$

Here, $g_m$ is the total membrane conductance ($g_m = \bar{g}_L + \bar{g}_{Nav}m^3h + \bar{g}_{Kv}n + \ldots$).

The point-dendrite model was populated with orientation-tuned synapses as described in the previous section. The orientation-tuned synaptic activity in the point dendrite was implemented by multiplying g$_{AMPA}$ and g$_{NMDA}$ of synapses with the average synaptic activity, w(θ), that depends on the orientation tuning of each synapse and the distribution of orientation preferences in the dendritic segment, according to **Fig 1d and 1e**. For each stimulation presynaptic event, all synapses were activated with a Poisson distributed delay (λ = 80 ms). The orientation tuning of the individual synapses was achieved by scaling the AMPA and NMDA conductance according to the synapse's tuning curve given by a half-circular normal distribution with σ$_{tuning}$ = 11˚ (see **Fig 1c**) [42,55,72,108]. To create a stimulation train consisting of several input events, we replicated the synaptic activation 2 times with a constant delay of 300 ms in between events. Based on our analysis in the Methods section "K⁺ diffusion state," the **S2 Text**, and the **S9–S13 Figs**, we modeled the activity-dependent local increase in [K⁺]$_o$ a step function in between events: The first stimulation event was regarded as the control condition with $\Delta E_K^+$ = 0 mV, and the 2 subsequent events as the experimental condition for a given $E_K^+$ shift in the interval $\Delta E_K^+ \in$ [6:18 mV]. We used a V$_m$ of –30 mV as a threshold criterion for identifying dendritic spikes. This value was chosen because we observed the largest change in V$_m$ around this value, as synaptic integration transitions from subthreshold input summation to suprathreshold spiking. To evaluate the effect of nonspecific inhibitory (GABA$_A$) input, we repeated our point-dendrite model simulations in the presence of inhibition. Specifically, we simulated the activation of GABA$_A$ synapses randomly activated at 10 Hz (**S2 Fig**).

## Linear dendritic spike function

To investigate the relationship between the average synaptic activity, w(θ), (**Fig 1e**), the $\Delta E_K^+$, and the number of synapses (N) for the emergence of a dendritic spike, we independently varied each variable in small increments in the point-dendrite model. By doing so, we identified the lowest value of $\Delta E_K^+$ that was able to push the V$_m$ over the threshold of –30 mV, our working criterion for dendritic spike generation, given N and w. We assumed that the relation between the 3 parameters was linear, and we, therefore, fitted a plane to obtain a model of $\Delta E_K^+$ needed to trigger dendritic spikes as a function of N and w. For this, we used the planar equation αw + βN + ν = 0 and solved it as Ax = b (**S3** and **S4 Figs** and Table 1 for coefficients):

$$\begin{pmatrix} \alpha \\ \beta \\ \nu \end{pmatrix} = (AA^T)^{-1}A^T b \tag{14}$$

From this equation, we can see that adding 1 additional synapse would lower the required $\Delta E_K^+$ for a dendritic spike by 2.08 mV. Accordingly, a 10% increase in w would lower the

**Table 1. Fitted coefficients for dendritic spike emergence seen in S3 Fig.**

| Coeff. | α | β | ν |
| --- | --- | --- | --- |
| Fitted value | –38.43 | –2.08 | 52.32 |

required $\Delta E_{K}^{+}$ for a dendritic spike by 3.8 mV. Note that in this work w varies between 0 and 1 and the maximum number of synapses on a dendritic segment is 13 [51–53]. A combination of sufficiently high number of synapses N and w would lead to negative $\Delta E_{K}^{+}$, in which case we assumed to correspond to no $E_{K}^{+}$ shift. From this, we could then estimate the dendritic spike probability as a function of $\Delta E_{K}^{+}$ and stimulus orientation. Using statistical formalism for the setup of the dendritic segments with similarly tuned inputs, we sampled the number of synapses and synaptic orientation preferences of $10^4$ segments, and calculated the average synaptic activity, w(θ). Following, we determined the $\Delta E_{K}^{+}$ needed to generate a dendritic spike according to $\Delta E_{K}^{+}$ (N, w) = αN + βw + ν for each stimulus orientation. For a stimulus presented in the target orientation, we compared this value with a set of $\Delta E_{K}^{+}$ values in [0, 6, 10, 18 mV] to determine whether each sampled segment fired a dendritic spike or not. To obtain the set $\Delta E_{K}^{+}$ values for the other orientations, we multiplied the set $\Delta E_{K}^{+}$ values of the target orientation with the normalized version of the synaptic activity factor seen in **Fig 1f** and repeated the comparison with the respective needed $\Delta E_{K}^{+}$ value. Finally, we calculated the fraction of dendritic segments, at each orientation, that would be expected to show dendritic spikes and converted this into a probability.

## Current-voltage attractor landscape

To understand the biophysical principles governing the regulation of dendritic spikes by local K$^+$ changes, we used dynamical systems theory [74]. While our point-dendrite model is defined by the differential Eq 10, it is not possible to solve it analytically. Rather, we chose to plot the exact solutions to the differential equations evaluated at different $V_m$ as a vector plot, which in essence is similar to summing all I-V curves from the included intrinsic and synaptic channels. We thus have $V_m$ on the first axis, and on the second we have its derivative which in our case is denoted $\frac{dV_m}{dt} = -\frac{I}{C}$. The time evolution of the synaptic currents in the point dendrite model was described as before (see S1 Text). Synaptic receptors were simulated in the time interval [0: 100 ms] using the Euler method. For each time step, an I-V curve was generated and the resulting I-V curves were concatenated to create a time-dependent I-V landscape (**Fig 3e** and **S1 Video**). To compare the effect of [K$^+$]$_o$ changes, we simulated the system with $\Delta E_{K}^{+}$ = 0 mV and $\Delta E_{K}^{+}$ = 18 mV.

## Abstract neuron model

Using the NEURON simulation environment, we constructed an abstract neuron model to systematically test the role of local [K$^+$]$_o$ elevations on dendritic input integration and somatic output firing. The dendrite's morphology was constructed as a fractal tree (**Fig 4a**): for each dendritic section generation, 3 new sections, each with half the length of the former, were added (length of the most distant section: 20 μm). This gave a fractal with dimension D = ln 3/ln 2 = 1.58, mimicking the generic compartmentalization and fractal dimensionality of the apical tree of pyramidal neurons [75,76]. The fractal dendritic tree was extended with a trunk dendritic compartment (500 μm) and a soma (length 30 μm, diameter 10 μm) to complete the neuron morphology, and intrinsic ion channels (K$_{leak}$, Na$_P$, Na$_T$, K$_P$, K$_T$, K$_{DR}$, K$_{Ca}$, KM, Ca$_{LVA}$, Ca$_{HVA}$, and HCN) were inserted into the cellular compartments (**S2 Table**), based on [73]. Using this morphology enabled us to test the effect without hidden edge cases caused by morphology, as regardless of the placement of the cluster the effect should remain the same. The fractal dimension kept the properties of the neuron as close as possible to that of a detailed morphology.

To investigate the compartmentalization of dendritic integration under the different $\Delta E_{K}^{+}$ values, we stimulated synapses residing in segments with similar orientation tuning on the

most distant dendritic branches [38,77,78]. For this, a Poisson-distributed number of dendritic segments ($\lambda$ = 32 segments) were selected to host the similarly tuned inputs in each simulation iteration, and the number of synapses and their orientation tuning were sampled as described above. Only distal dendrites received direct input, to ensure a constant distance to the soma. The local $\Delta E_K^+$ of stimulated dendrites were grouped into 3 conditions: no shift ($\Delta E_K^+$ = 0 mV), small shifts ($\Delta E_K^+$ = 6 mV), or large shifts ($\Delta E_K^+$ = 18 mV), based on the $[K^+]_o$ changes found in **Fig 1**. As before, the orientation tuning of individual synapses was achieved by scaling the AMPA and NMDA conductance. To simulate a realistic somatic firing pattern, the synapses within each dendritic segment were activated randomly given by a Poisson distribution with $\lambda$ = 30 ms. This stimulation protocol was repeated 3 times with a 300 ms delay between stimulation events. Each neuron setup was simulated with each of the 3 $\Delta E_K^+$ conditions to directly assess the effect of the local K$^+$ changes.

## Multiple ionic contributions

In this investigation, we only considered the activity-induced changes of $[K^+]_o$ and assumed the concentration of all other ions to remain constant. We assume contributions from Na$^+$ and Ca$^{2+}$ to be minimal and their effects covered within the range of the $E_K^+$ demonstrated. This difference in the respective contributions arises from the ratio between the intra and extracellular volume, where the former is assumed to be at least a factor of 2 larger [62–64] (the size of the extracellular space ranges from 0.04 to 0.5 μm [63,104]), and is smaller than the pyramidal neuron dendritic diameter [105,106] and is also motivated by our previous experimental recordings documenting in vivo elevation of $[K^+]_o$ [14]. This results in greater changes seen in the extracellular space, and combined with the formulation of the Nernst equation, the shift in $E_K^+$ outweighs the other ions. For a more in-depth discussion about this, see **S3 Text** and **S14 Fig**.

## Neuronal activity measures

To obtain a better understanding of how the local $[K^+]_o$ changes impact the activity of the neuronal model we introduced 2 additional measures, in addition to the conventional measurement of somatic firing rate: Neuronal firing gain transformation and area under the curve (AUC) for the integrated dendritic $V_m$ signal.

## Neuronal firing gain transformation

We assumed that the gain modulation of the somatic firing tuning curve could be described by a multiplicative or additive function, similar to previously described in the visual cortex of mice [79]. The multiplicative and additive functions are given by:

$$FR_{After}(\theta) = \xi_{mul}\, FR_{Before}(\theta) \tag{15}$$

$$FR_{After}(\theta) = \xi_{add}\, FR_{Before}(\theta) \tag{16}$$

Here, $FR_{Before}(\theta)$ and $FR_{After}(\theta)$ are the somatic firing rates at a given orientation with non-shifted or shifted $E_K^+$ values, respectively, $\xi$ is the fit parameter that describes the change in firing rate either as a gain coefficient or as a gain constant for the multiplicative and additive gain transformation, respectively. To determine the fit parameters, we fitted the 2 functions to the firing rate tuning curve data using the $\chi^2$ fitting method (**Table 2** and **S7 Fig**). Only the multiplicative method was able to produce a satisfying fit and capture the main features of the

**Table 2. Fitted gain parameters.** Gain parameters obtained for the multiplicative (top) and additive functions (bottom) for the small and large $E_K^+$ shifts (left and right, respectively). Errors were assumed Gaussian and reported as standard deviations. See also **S7 Fig**.

| | $^\xi$ **Small** $\Delta E_K^+$ | $^\xi$ **Large** $\Delta E_K^+$ |
|---|---|---|
| Multiplicative | 1.64 ± 0.21 | 2.73 ± 0.32 |
| Additive | 2.73 ± 0.75 | 7.06 ± 0.33 |

tuning curve, whereas the additive only captured the responses to orientations close to the target orientation.

## Area under the curve

To understand how dendritic $E_K^+$ shifts shape dendritic gain, we computed the AUC of the $V_m$ measured at the base of the dendritic trunk, over an interval of 500 ms to capture the full synaptic activation during an event. To exclude the contribution of bAPs, we here turned off the voltage-gated sodium channels in the soma and trunk, which eliminated somatic firing. To calculate the AUC, we first subtracted the $V_m$ baseline of the control signal, corresponding to $\Delta E_K^+ = 0$ mV, from all traces, and used the trapezoidal method.

## Orientation selectivity index

The orientation selectivity index was computed as 1 –circular variance (CV) in orientation space given by:

$$Orientation\ selectivity\ index = 1 - |CV| = \left| \sum_k r_k \frac{e^{-i2\theta_k}}{\sum_k r_k} \right| \qquad (17)$$

Where $r_k$ is a measurable output at orientation $\theta_k$ (in radians); here, we used dendritic spiking probability and somatic firing rate as output measures. To compute OSI, we mirrored the results obtained in the simulation between $[0:90°]$ to obtain values in the interval $[-90:90°]$.

## Statistics

To determine if somatic firing rates, obtained with either small or large local $E_K^+$ shifts, differed from firing rates obtained with no $E_K^+$ shift, we performed one-tailed Student's $t$ tests on the residuals of the somatic firing rates. $P < 0.05$ was considered statistically significant, where $^*P < 0.05$, $^{**}P < 0.01$, and $^{***}P < 0.001$.

## Supporting information

**S1 Text. Biophysical model setup.**
(PDF)

**S2 Text. Spatial and temporal properties of K$^+$ diffusion.**
(PDF)

**S3 Text. Electrochemical potential shifts of ions.**
(PDF)

**S1 Fig. Voltage response of the point dendrite with excitatory inputs.** Example dendrite $V_m$ traces for a similarly tuned dendritic segment as a function of stimulus orientation relative to target orientation. Individual trials are in gray and average is in teal. The colored segments in the inset (right) show the impact of the $E_K^+$ shift (comparison of the first and third responses).

The first stimulation event induces a small $E_K^+$ shift (10 mV) for the target orientation. For the rest of stimulus orientations, the shift in $E_K^+$ is scaled according to **Fig 1f**. $E_K^+$ shifts increase dendritic spike occurrence and dendritic spike duration.
(PDF)

**S2 Fig. Voltage response of the point dendrite with excitatory and inhibitory inputs.** Top 2 plots: Example dendrite $V_m$ traces for different orientations, as in S1 Fig, when GABA$_A$ synapses (0.2 synapses/μm density) were randomly stimulated with mean Poisson frequency 10 Hz. Following plots show example dendrite $V_m$ traces for target orientation with increasing the inhibitory conductance and mean Poisson frequency of inhibitory activation. In each subplot with gray are voltage responses from 10 repetitions, and in teal is the mean response. The colored segments in the inset show the impact of the $E_K^+$ shift (comparison of the first and third responses). As in S1 Fig, the first stimulation event induces a small $E_K^+$ shift (10 mV) for the target orientation. High $E_K^+$ shifts increase dendritic spike occurrence and dendritic spike duration in the presence of nonspecific inhibition. High enough inhibition (bottom panel) reduces dendritic spike occurrence irrespective of the $E_K^+$ shift.
(PDF)

**S3 Fig. Dendritic spike emergence plane fit.** We created multiple input–output curves by varying the average synaptic activity (w), the number of synapses on the dendritic segment (N), and $\Delta E_K^+$, in the point-neuron model, while having dendritic spike occurrence ($V_m$ above −30 mV) as the output measure. This was done multiple times for each parameter combination and the resulting scatter of points in 3D space is shown here (blue circles). Following, we fitted a plane to this data (black grid), as we assumed the relation between w, N, $\Delta E_K^+$ and dendritic spike generation to be linear; points on the plane denote the parameter sets which were able to generate a dendritic spike.
(PDF)

**S4 Fig. Dendritic spike emergence for constant N.** Data similar to S3 Fig, presented here as a scatter plot for different numbers of synapses, N (colors). As suggested by the planar plot, the relationship between $\Delta E_K^+$ and w is well approximated by a linear relationship. In addition, the slope of each line increases linearly with the number of synapses.
(PDF)

**S5 Fig. Synaptic activity and number of synapses regulate active dendritic properties.** Heat map showing peak $V_m$ depolarization as a function of mean synaptic activity (w) and the number of synapses (N) in the point dendrite model. Dotted lines indicate the transition to dendritic spiking. Computed for $\Delta E_K^+ = 0$ mV. Compare this figure to Fig 2f of main text.
(PDF)

**S6 Fig. Orientation tuning of a layer 5 pyramidal cortical neuron, at different $\Delta E K_K^+$.** Somatic firing rate tuning curve for different dendritic $E_K^+$ shift magnitudes, similar to Fig 4. To further verify the results obtained with the abstract neuron of Fig 4, we simulated a neuron with detailed, reconstructed morphology. Here, we chose the widely used model of a layer 5 pyramidal neuron, obtained from [1]. The neuronal setup was kept similar to the one used for the abstract neuron, with active conductances listed in S3 Table. Synapses with similar orientation preferences were on the tips of the apical dendrites, similar to the procedure for the abstract neuron model in the main text. As before, the orientation tuning of individual synapses was achieved by scaling the AMPA and NMDA currents according to the synapse's tuning curve. Apical segments that experienced input stimulus underwent a shift in $E_K^+$. The local $\Delta E_K^+$ imposed during simulations were grouped into 3 conditions: no shift ($\Delta E_K^+ = 0$ mV),

small shifts ($\Delta E_K^+$ = 6 mV), or large shifts ($\Delta E_K^+$ = 18 mV), based on the $[K^+]_o$ changes found in Fig 1. To simulate a realistic somatic firing pattern, the synapses within each dendritic segment were activated randomly given by a Poisson distribution with λ = 30 ms. This stimulation protocol was repeated 3 times with a 300 ms delay between stimulation events. Each neuron setup was simulated with each of the 3 $\Delta E_K^+$ conditions to directly assess the effect of the local $E_K^+$ changes.
(PDF)

**S7 Fig. Gain function fit.** Left, multiplicative and additive gain function predictions plotted together with the original somatic firing rate for the low $\Delta E_K^+$ condition as a function of stimulus orientation relative to target orientation. The ξ values denote the fit parameter that describes the firing rate change either as a gain coefficient or as a gain constant for the multiplicative ($\xi_{mul}$) and additive ($\xi_{add}$) gain transformation, respectively. Predictions were generated by either adding $\xi_{add}$ or multiplying $\xi_{mul}$ to the tuning curve with no $E_K^+$ change. Errors were assumed Gaussian and reported as ± standard deviations. Right, actual multiplicative and additive gain function fits plotted with the simulated somatic firing rate data with no $E_K^+$ shift (firing rate before) against low $E_K^+$ shift (firing rate after). The original firing rate data was plotted as ± standard deviations on each axis. To fit the data, we have only considered data with nonzero standard deviation.
(PDF)

**S8 Fig. Orientation tuning curve of the abstract neuron model with varying synaptic strength.** Somatic tuning curve for different values of synaptic input strength while $g_{AMPA}$ and $g_{NMDA}$ was varied in the interval ±5% relative to the strength used in the simulations of Fig 4c. For these simulations $\Delta E_K^+$ = 0 mV.
(PDF)

**S9 Fig. Discretization of the extracellular space surrounding serially connected dendrite segments.** (a) The 10 μm dendritic segments with diameter $R_1$ = 1 μm were concatenated to create 1 dendrite with total length 110 μm. The surface of the dendrite is unfolded to create a 2D surface where we simulate the movement of K$^+$. The segments were either populated with similarly or diversely tuned synapses. Colored points indicate the placement of synapses on a 2D grid around a dendritic segment, and their color represents their target orientation, similar to Fig 1C. (b) Representation of K$^+$ efflux from one synapse at different orientations. If a synapse is activated close to its preferred orientation (red line), the total amount of K$^+$ efflux will be higher than a non-preferred stimulus (black line). Note that tpeak corresponds to the peak of K$^+$ efflux, occurring after synaptic activation.
(PDF)

**S10 Fig. Dynamics of $\Delta E_K^+$ \with varying stimulus orientation.** (a) Top: Example $\Delta E_K^+$ traces over time for a stimulus presented at the target orientation, arbitrary set at 0˚. Solid lines show the evolution of $\Delta E_K^+$ of each dendritic segment along the dendrite (different colors), with the central dendritic segment, receiving similarly tuned synapses, being the reference point (distance = 0 μm, light orange). Dotted green line shows $\Delta E_K^+$ over time of a dendrite receiving exclusively diversely tuned input. After synaptic activation, the peak $\Delta E_K^+$ is reached within 150–200 ms. As also shown in Fig 1, the largest shift in $E_K^+$ is seen for the segment receiving similarly tuned synapses. Neighboring segments, receiving diversely tuned synapses, display a gradually decreasing shift, and for dendritic segments in distance, >40 μm, the $\Delta E_K^+$ is similar to the one expected from a dendrite receiving exclusively diversely tuned synapses (dotted green line). Middle, bottom: $\Delta E_K^+$ as function of stimulus orientation (22.5˚ and 45˚, respectively). For a stimulus orientation far from target orientation, the shift in the $E_K^+$

becomes smaller and similar to the diverse input tuning regime, as per Fig 1g and 1h. Overall, for the diversely tuned segments the smooth $\Delta E_K{}^+$ is as a result well-mixed and uniform $[K^+]_o$ throughout the outer cylinder. For the similarly tuned segment, an initial drop before stabilizing in $\Delta E_K{}^+$ is due to the concentration gradients with its adjacent segments, still maintaining higher $\Delta E_K{}^+$ levels when compared to diversely tuned segments. (b) Same data as above, for $\Delta[K^+]_o$. $\Delta[K^+]_o$ lies within the interval [1:5 mM], as per Fig 1, and follows the similar trend to the $\Delta E_K{}^+$. For all plots, bottom rows indicate the activation timings of individual synapses. (PDF)

**S11 Fig. Diffusion in the intracellular space does not change the dynamics of $\Delta E_K{}^+$.** Example $\Delta E_K{}^+$ traces over time for a stimulus presented at the target orientation, at different distances from the dendritic segment receiving similarly tuned synapses. Colors are as per S10 Fig Solid line: $E_K{}^+$ calculated by taking into account changes both in the intracellular and extracellular $[K^+]$. Dotted line: $E_K{}^+$ calculated by changes in the $[K^+]_o$ while $[K^+]_i$ is constant. Only subtle differences are noted, with the full model laying just below the model using only $[K^+]_o$. This is to be expected, as a reduction of $[K^+]_i$ would lower $E_K{}^+$. The minimal effect of $[K^+]_i$ changes is due to the differences in the extracellular and intracellular spaces sizes, described by $V_R$. (PDF)

**S12 Fig. Dynamics of $\Delta E_K{}^+$ with varying $K_{dec}$.** Example $\Delta E_K{}^+$ traces over time for a stimulus presented at the target orientation with varying the strength of the potassium pumps (top: $K_{dec}$ = $1.9 \cdot 10^{-8}$ m/s, middle: $K_{dec}$ = $2.9 \cdot 10^{-8}$ m/s, bottom: $K_{dec}$ = $3.9 \cdot 10^{-8}$ m/s). Increasing the decay constant of the $Na^+/K^+$ pump lowers the $\Delta E_K{}^+$ both for the similarly and diversely tuned segments and reduces the differences in $\Delta E_K{}^+$ between the 2 types over time. (PDF)

**S13 Fig. Dynamics of $\Delta E_K{}^+$ with varying the interstimulus interval.** Example $\Delta E_K{}^+$ traces over time for a stimulus presented at the target orientation for interstimulus intervals 200 ms (top), 300 ms (middle), and 400 ms (bottom). For all segments, the shorter interstimulus interval (200 ms) allows for the temporal summation of the extracellular $[K^+]$ yet reaches similar levels of $\Delta E_K{}^+$ when compared to longer intervals. (PDF)

**S14 Fig. Electrochemical potential change of ions for different $\Delta[ion]$.** Left: Absolute changes in the reverse potential for $K^+$ and $Na^+$ for different $V_R$ and $\Delta[ion]$. Center: Ratio between the change in the potential for $K^+$ and $Na^+$. Right: Absolute changes in the potential for $Ca^{2+}$ ions at different $V_R$ and $\Delta[ion]$. Note that the range of the first axis is in μm and not directly comparable with the left plot axis. (PDF)

**S15 Fig. Frequency-based modulation of responses.** Example $\Delta E_K{}^+$ traces over time for stimuli presented at 0° (top), 22.5° (middle), and 45° (bottom) following a single stimulation event. Solid lines show the data for weight-modulated orientation tuning of synapses, i.e., based on $w_{syn(i,j)}(\theta)$, with each synapse's activation time drawn from a Poisson distribution with $\lambda$ = 80 ms, as per **S10–S13 Figs**. Dotted lines show the data for frequency-modulated orientation tuning of synapses, i.e., when each synapse is activated by a Poisson train whose frequency depends on the synapse's orientation preference (see **S2 Text** for details). Timing of inputs is plotted as raster plots (black lines: weight-modulated, gray lines: frequency-modulated). Following the stimulation event, the simulation runs for 2 s to show the return to baseline ($\Delta E_K{}^+$ = 0 mV). The 2 approaches display similar dynamics, reaching same peak $\Delta E_K{}^+$

responses for the similarly tuned segment within $\sim 200$ ms. $\Delta E_{\mathrm{K}}^{+}$ is modulated by the stimulus orientation mainly in the segment receiving similarly tuned inputs. In contrast, the diversely tuned segments sustain a fixed $\Delta E_{\mathrm{K}}^{+}$ irrespective of the stimulus orientation, as per **Fig 1g**. Neighboring segments, receiving diversely tuned synapses, display a sharper decrease in the $E_{\mathrm{K}}^{+}$ shift for the frequency-modulated input compared to the weight-modulated input. The slow decay to the steady state $E_{\mathrm{K}}^{+}$ indicates the time window within which a second stimulation would be modulated by the elevated $E_{\mathrm{K}}^{+}$. Colors of plotted lines are as **S10 Fig**. (PDF)

**S1 Table. Active conductances of the point-dendrite model.**
(PDF)

**S2 Table. Active conductances of the abstract neuron model.**
(PDF)

**S3 Table. Active conductances of the neuron model for the L5 PC neuron.**
(PDF)

**S1 Video. Current-voltage attractor landscape during a dendritic spike.** (a) I-V curves including intrinsic ion channels and NMDA receptors during the generation of a dendritic NMDA spike without and with $E_{\mathrm{K}}^{+}$ shift. Solid and open points indicate stable and unstable fixed points, respectively, and arrows indicate system flow direction. (b) Simplified outline of the approximate $V_{\mathrm{m}}$ during the dendritic NMDA spike with and without $E_{\mathrm{K}}^{+}$ shift, highlighting the voltage levels the $V_{\mathrm{m}}$ is attracted toward in a. (c) Heat maps showing the temporal evolution of the I-V curves without and with $E_{\mathrm{K}}^{+}$ shift. The red line indicates the corresponding I-V curves shown in a, full and dotted white lines indicate stable and unstable fixed points, respectively, and the magenta and green lines being drawn indicate the instantaneous $V_{\mathrm{m}}$ for the system without and with $E_{\mathrm{K}}^{+}$ shift, respectively.
(MP4)

## Acknowledgments

We thank Akihiro Matsumoto, Alessandra Lucchetti, Eva Maria Meier Carlsen, Ioannis Matthaiakakis, and Stamatios Aliprantis for discussions and comments on the manuscript.

## Author Contributions

**Conceptualization:** Malthe S. Nordentoft, Naoya Takahashi, Rune N. Rasmussen, Athanasia Papoutsi.

**Data curation:** Malthe S. Nordentoft.

**Formal analysis:** Malthe S. Nordentoft.

**Funding acquisition:** Naoya Takahashi, Mathias S. Heltberg, Mogens H. Jensen, Rune N. Rasmussen, Athanasia Papoutsi.

**Investigation:** Malthe S. Nordentoft.

**Methodology:** Malthe S. Nordentoft.

**Project administration:** Malthe S. Nordentoft, Naoya Takahashi, Mathias S. Heltberg, Mogens H. Jensen, Rune N. Rasmussen, Athanasia Papoutsi.

**Software:** Malthe S. Nordentoft.

**Supervision:** Rune N. Rasmussen, Athanasia Papoutsi.

**Validation:** Mathias S. Heltberg, Mogens H. Jensen, Athanasia Papoutsi.

**Visualization:** Malthe S. Nordentoft, Rune N. Rasmussen.

**Writing – original draft:** Malthe S. Nordentoft, Naoya Takahashi, Rune N. Rasmussen, Athanasia Papoutsi.

**Writing – review & editing:** Malthe S. Nordentoft, Naoya Takahashi, Rune N. Rasmussen, Athanasia Papoutsi.

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
