## [Editor Report · Decision Letter 0]

21 Mar 2024

Dear Nassi, 

Thank you for submitting your manuscript entitled "Local changes in potassium ions regulates input integration in active dendrites" for consideration as a Research Article by PLOS Biology.

Your manuscript has now been evaluated by the PLOS Biology editorial staff and I am writing to let you know that we would like to send your submission out for external peer review. Please note that we have not been able to get advice from an Academic Editor so far and, therefore, are yet to make a firm call on the conceptual advance of your study. We will discuss this aspect with an academic editor once we have received the reviewer reports and will be looking for strong enthusiasm from the reviewers. 

Once your full submission is complete, your paper will undergo a series of checks in preparation for peer review. After your manuscript has passed the checks it will be sent out for review. To provide the metadata for your submission, please Login to Editorial Manager (https://www.editorialmanager.com/pbiology) within two working days, i.e. by Mar 23 2024 11:59PM.

Kind regards,

Christian

Christian Schnell, PhD

Senior Editor

PLOS Biology

cschnell@plos.org

---

## [Decision Letter · Decision Letter 1]

3 May 2024

Dear Nassi,

Thank you for your patience while your manuscript "Local changes in potassium ions regulates input integration in active dendrites" was peer-reviewed at PLOS Biology. It has now been evaluated by the PLOS Biology editors, an Academic Editor with relevant expertise, and by several independent reviewers. 

In light of the reviews, which you will find at the end of this email, we would like to invite you to revise the work to thoroughly address the reviewers' reports.

As you will see below, the reviewers think that the study is overall well executed and provides important insights. Reviewer 1 has concerns about not including spatial effects of potassium concentrations and differences between neighboring dendritic segments. Reviewer 2 lists a number of specific questions and comments, that will also require some additional analyses and possibly an extension of the model, but most of them can probably be addressed through textual revisions.

Given the extent of revision needed, we cannot make a decision about publication until we have seen the revised manuscript and your response to the reviewers' comments. Your revised manuscript is likely to be sent for further evaluation by all or a subset of the reviewers.

**IMPORTANT - SUBMITTING YOUR REVISION**

*Re-submission Checklist*

*Published Peer Review*

*PLOS Data Policy*

*Blot and Gel Data Policy*

Sincerely,

Christian

Christian Schnell, PhD

Senior Editor

PLOS Biology

cschnell@plos.org

REVIEWS:

Reviewer #1: The manuscript of Nordentoft et al. investigates the effect of extracellular potassium concentration changes on dendritic excitability and neuronal responses in abstract visual cortical neuron models. IUsing they modeling approach they argue that 1) the [K]_o can substantially change during intense stimulation, especially in dendritic segments receiving clustered inputs; 2) The elevated extra- (and reduced intra-)cellular potassium concentration changes the reversal potential and reduces the driving force of all potassium currents in the membrane; 3) As a consequence of the reduced driving force for K, the excitability of the dendritic tree increases locally, including elongation of the NMDA spikes and increasing gain of the inputs. Importantly, the authors claim that the stongest effect can be observed when the neurons receives strong, clustered input, but even in this case, the extracellular concentration changes remain localized.

The effects are clear, and the proposed mechanism seems to be an important biophysical process potentially regulating dendritic excitability locally. However, I'm not convinced that all the spatial aspects of the changes in the potassium concentration can be neglected and that we can expect a large, but selective differences in the [K]_o experienced by neighbouring dendritic segments.

I think that this is really the most important assumption of the paper: how much can we believe that the changes in [K]_outside (and [K]_inside) remain localized? There are several important factors potentially influenceing [K]_outside not considered here:

- First, lateral diffusion. They argue in the Methods (K+ diffusion state), that within the 10um segment they study, the [K]_o reaches a spatially uniform concentration within 300 ms. At steady state after stimulation, the [K]_o can be twice as large as during rest. They also argue, that different segments are well isolated from each other even on this larger time scale, since on avrage, K+ ions only move ≈30 μm during a period of 600 ms. However, these arguments are not supported by exact calculations or simulations, just back of the envelop-type estimations. 

I acknowedge that the fine details of many of these processes are unknown, and thus building a full model of extracellular K+ dynamics is unrealistic and beyond the scope of the current paper. However, I believe that a simple simulation of the K+ diffusion along a dendritic branch could convincinly demonstrate the spatial and temporal scales of the K+ concentration changes. Specifically, I would perform a simulation where 1. a synapse cluster (7-13 spines) is placed on the middle of a ~100 μm long dendritic segment. 2. Further synapses are added with similar density but uniform tuning. 3. Synapses are stimulated with rate-modulated Poisson spike train according to a slowly changing stimulus orientation. 4. The synaptic and intrinsic currents of the dendrite are recorded and the diffusion of K+ is simulated in the extra- and intracellular space. 5. K+ reversal potential is calculated from the local concentration. 6. K/Na exchanger is also simulated to maintain the long-term balance of K+ concentration. 

- Second, the K+ concentration at a given segment is influenced by K+ release from neighbouring branches, that belong to the same or a different neuron. This effect is probably hard to consider exactly, but given the high density of dendritic membranes and synapses it may have an important contribution. Estimating the number of synapses within a 5um diameter sphere centered on a particular branch gives ~500 synapses (using 1.3 synapse / um3 from https://doi.org/10.1371/journal.pone.0198131). Only 2% of these belongs to the same cluster. Is it sure that we can ignore the rest? Please discuss this point.

Specific comments:

Figure 1: panel labels are CAPITAL letters, references are small. 

Fig 2c: does the resting potential also change after the elevation of the [K]_o? Is this responsible for the shift of the curves before stimulation?

Page 6: "we also included a similar test using a reconstructed neuron morphology based on a previously published model, see (Supplementary Figure 4)." -> SF5.

Reviewer #2: Nordentoft and colleagues explore the role of the transient changes in intra- and extracellular potassium concentrations for dendritic (and neuronal) integration of synaptic inputs. Based on a dendritic model and a simplified neuronal model with more realistic morphological structure, the authors show how co-tuned excitatory inputs may elicit stronger outputs when changes in the reversal potential of potassium channels are taken into account. Complementary, the authors show that dynamically controlling this reversal potential can effectively modulate the input/output function of both dendrite (local membrane potential) and neuron (output spikes), revealing an interesting new role for potassium channels. To the best of my knowledge, this is a novel approach to elucidate the role of ionic concentrations (and their dynamics) for neuronal processing. I really like the approach and appreciate the modelling of both analytical, point-dendrite, and morphological neurons. In my view, the results are novel and relevant, and would be of interest to the broad neuroscience community.

However, some points should be addressed by the authors to clarify the origins of some of the phenomena presented in their manuscript. For example, the model used by the authors comprises four different potassium channels with distinct dynamics described by activating and inactivating gates, and the role of each of these channels for the modulation on the input/output function is not clear. Additionally, it's not clear whether the effect reported by the authors could also be achieved by strengthening synapses. I elaborate on this and other major and minor points below.

Major

1) Focus on the visual cortex. As I was reading the manuscript I would often think that the author's approach and results would fit very well the description of many other brain regions rather than visual cortex only which is only mentioned briefly in the discussion section. I think that the manuscript would benefit greatly if the model and results were built in a more general framework from the beginning, using the visual cortex as one of many examples given that other cortices (and even hippocampus if place fields are considered) have, in general, tuning curves. 

2) Lack of inhibitory synapses. Until the discussion section, the authors refer to the connections in their model simply as "synapses", without specifying their types, and I think it would be important to clarify already in the introduction and beginning of results section that they are only considering the activation of excitatory synapses. Additionally, I think that a more detailed description of the rationale for not including inhibitory connections is lacking. Why was it not included? Is it because another set of free parameters would have to be explored, which prevents a more detailed description of the phenomena? Would it be possible to consider at least one case in which inhibitory inputs are unspecific?

3) Extracellular compared to intracellular volume. The authors consider a fairly small extracellular volume compared to the intracellular one (e.g., Fig. 1g,h) and refer to previous work for this choice. However, as far as I understood, Tønnesen et al. (2018) have shown that the intracellular volume is bigger than the extracellular volume when considering a region with many somas (and adding up all the different neurons). For a single (and small) dendritic compartment (far from the soma), I would expect a much larger extracellular volume compared to the intracellular one. Considering that the extracellular volume (defined by a cylinder with radius R2 and length L) is a free parameter of the model, I would like to suggest checking the values of Δ[K+]o and Δ[K+]i, and consequently ΔEK+, for two extreme conditions: infinite and zero extracellular volumes.

4) Role of individual channels. The authors implement four different potassium channels in their models, each one with distinct activation and inactivation functions according to the Supplementary Material. First, it would be interesting to discuss whether changes in molar concentrations may have an effect in the activation and inactivation functions of these channels - is there any reference for that? Second, I think it would be important to pinpoint the effect of each ion channel. One question that naturally arises is whether the seemingly independence of changes in ΔEK+ on volume ratio (Supplementary Fig. 4) is due to the combination of the four ion channels or not. This could, e.g., explain the discrepancy with sodium and calcium channels, which seem to follow very similar trends (Supplementary Fig. 4).

5) Dynamics of EK+. One of the main assumptions is that "1) Local increases in [K+]o are proportional to synaptic activity level." If that's the case, I would interpret that ΔEK+ is dynamic and follows the activation of AMPA/NMDA channels (maybe with some slow time constant). For the statistical analysis, assumption "3) [K+]o change is stable in space and time along the short dendritic segment." is a valid (and necessary) one, but for the dendritic dynamics this assumption might be too strong. From my understanding, both point-dendrite and morphological neuron were implemented with fixed EK+ for different dendritic segments based on whether the connections onto them are co-tuned or not. I find it to be a very good initial assessment of the dendritic response, but it is ultimately at odds with assumption (1) because inputs are not constant (allowing for definition of co-tuned), which means that ΔEK+ should increase when the target orientation is active, and go to zero otherwise. In Fig. 2c, it's clear that the baseline Vm (when no input is present) is different for different ΔEK+ and, according to assumption (1), they should all start from the same resting state if other orientations were active beforehand. Is there strong evidence for the target orientation being activated consistently with a maximum of 300 ms delay? What would happen if ΔEK+ is dynamic?

6) Functional consequences. The authors explore the functional consequences of changes in local [K+]o by the modulation in postsynaptic firing. These results seem to align very well with the dendritic spike probability (comparing Figs. 2g and 4c) which leads me to the following question. Is the modulated postsynaptic response for greater dendritic distances to the soma (Fig. 4g) solely a consequence of the changes in potassium concentration throughout the dendritic tree (as stated by the authors) or a consequence of a greater local response? In summary, is the postsynaptic response the same for (i) strong synapses and no potassium modulation and (ii) weak synapses and potassium modulation? This also leads to another question regarding synaptic weights below.

7) Parameter set. Eq. 2 describes the approximated combination of parameters ΔEK+, N, and w, namely, change in potassium reversal potential, number of synapses, and synaptic strength, that allow the local dendrite to trigger dendritic spikes. The authors found a linear relationship for these parameters, which intuitively would mean that distinct combinations of these parameters would give rise to the same results. In this case, what would be the resulting plot in Fig. 2f for different synaptic weight values or number of synapses? If the x-axis of Fig. 2f is synaptic weight, w, would it look equivalent? If that's the case, I would like to suggest elaborating on the benefits of having changes in the potassium reversal potential if, e.g, having strong weights would have the same final result (e.g., in Fig 4).

Minor

8) I would like to suggest to the authors to submit PDFs with lines numbered so it's easier to address specific points during the review process.

Introduction

9) Vm is used once in the Introduction, thus I suggest writing "membrane potential".

10) Page 2: "These dendritic spikes support the nonlinear…, which in turn boost the neuron's input-output function…" Do the authors mean "modulate the neuron's input-output function"? The output can be boosted, but the function can only be modulated (or another synonym for modulation).

11) Page 2: "We focus on the organization of synaptic inputs tuned…" These are excitatory synaptic inputs. I would like to suggest clarifying this point.

12) Page 2: "... synaptic inputs can attain substantially higher [K+]o changes … These [K+]o elevations …" In the first sentence the authors use "higher changes" and later "these elevations". There is a mismatch here.

Results

13) It was not clear from the first paragraphs how the model was initially implemented. It was more clear after reading the subsequent section which described the point-dendrite model. If I understood correctly, all results presented in Fig. 1 are not taken from the implementation of the membrane potential dynamics, but from a statistical analysis. I would like to suggest clarifying that initially no dynamics were included in the model. I think this is a very interesting and powerful way to analyse such a system, and as so, deserve to be highlighted.

14) Page 3: "... we implemented 10 μm segments …" Following the point above, the word "implementation" gave me the impression that the model being presented here had membrane potential dynamics. It was also not clear to me whether these were supposed to represent isolated segments (like in the pound-dendrite model) or part of a morphological neuron. I would like to suggest clarifying these points.

15) Page 4, Eq. 1: Isn't there a minus sign missing here? Positive changes outside should reflect negative changes inside and vice versa, so both deltas should have opposite signs.

16) Page 4: "... by converting the Δ[K+]o into shifts in EK+..." I would like to suggest adding the equation used to convert one into the other here for clarity, given that this is an important implementation throughout the paper.

17) Page 4: "Synaptic inputs were activated by delivering a stimulation train consisting of three events…" The precise implementation was not clear to me. The equations for AMPA and NMDA channels in the Supplementary Material lack additional information for how these activity patterns were implemented. I would like to suggest a more detailed explanation in the Methods and/or Supplementary Material.

18) Page 5, Eq. 2: I could not find the parameter values for α, β, and ν. I would assume that β is negative - is it the case? Would that mean that ΔEK+ can be negative for very strong weights? How was w chosen?

19) Page 5: "The current barrier…" Isn't it the "voltage barrier"? The analysis of the system through fixed points is given by checking which values of Vm result in dVm/dt = 0, shown in Fig. 3. The neuron model is defined as dVm/dt = I, so that's why the analysis also works considering the current on the y-axis, but the definition of a fixed point is for dVm/dt. Same for the sentence "... it lowers the current barrier…" Isn't it "voltage barrier" here as well?

20) Page 6: "... segments of the dendritic tree randomly underwent EK+ shifts …" Why is it random? One of the assumptions to build the model according to the authors is "1) Local increases in [K+]o are proportional to synaptic activity level.". Following this argument, changes in EK+ should not be random, but proportional to the AMPA/NMDA activation in each dendrite. I would like to suggest clarifying this point and including a more detailed implementation in the Methods/Supplementary Material.

21) Page 6: ".. during a synaptic stimulation burst." I would like to suggest including details of the implementation in the Methods/Supplementary Material, e.g., duration of burst, number of spikes, etc.

22) Page 6: "... due to the local gain of the synaptic input, the EK+-shifted regimed is attenuated less with distance." If synaptic strengths were stronger at these dendrites, would the effect be the same without shifting EK+? If that's the case, synaptic plasticity could be optimising the connections so that the effect is larger without the need of EK+ changes.

23) Page 6: "... attenuated with a factor L1/3". How was this value calculated?

24) Fig. 1: If I understood correctly, all results presented in this figure were calculated/estimated without simulating the membrane potential dynamics explicitly. For clarity I would like to suggest writing this explicitly in the caption so that it is easier for the reader to immediately understand the difference between, e.g., Figs. 1 and 2.

25) Fig. 1, panel c: If I understood correctly, panel c shows an integration of the inputs over a large period in time, and not a snapshot (otherwise the synapses would not represent different orientations). I would like to suggest clarifying this point in the caption.

26) Fig. 2, panel h: The y-axes are different for peak and orientation. Starting the orientation axis at zero while starting the peak axis at 0.4 visually amplifies the effect of the changes in peak spike probability. I think it would be more of a fair comparison to use the same axis range for both.

27) Fig. 3, arrows: The arrows in panels a-d are hardly visible. I would like to suggest either removing them or making them more visible. 

28) Fig. 3, current: For the analysis of fixed points, the usual approach is to compare dVm/dt against Vm. This way it's clear that: 1) when the line crosses zero, there is no change (it's a fixed-point); 2) when the line is above zero, changes are positive, so Vm is moving to the right; and 3) when the line is below zero, changes are negative, so Vm is moving to the left. I find it a bit confusing that the authors plotted negative values above positive ones, inverting the usual presentation of fixed-point analysis. I would like to suggest changing the plot, or clarifying this in the text and caption.

29) Fig. 4, panel d. Similarly to Fig. 2h: one y-axis starts at zero and the other above zero, highlighting one effect more than the other.

Discussion

30) Page 7: "First, the fine-scale arrangement of orientation-tuned synaptic inputs determines the magnitude of activity dependent [K+]o changes…" If I understood the model correctly, this was not a result, but an assumption used to build the model. I would like to suggest rephrasing this sentence to reflect this point.

Methods

31) Page 13: The two first sections, if I understood correctly, describe the model without any voltage dynamics. I would like to suggest clarifying this point.

32) Page 14, Eqs. 6 and 7: As for Eq. 1, isn't there a negative sign missing here?

33) Page 14: I would like to suggest putting Eq. 8 in the Results section as well to help the readers understand the implementation.

34) Page 15: I would like to suggest including all terms in Eq. 11 for completeness.

35) Page 15, Eq. 12: The total conductance should include the gating terms (n, m, h, etc). I would like to suggest clarifying this point.

36) Page 15: "The orientation tuning… as we have done previously." I would like to suggest expanding on the details of the implementation for completeness. 

37) Page 15: "... mean synaptic activity (w…" The variable w is not defined for the implementation of the point-dendritic model nor the morphological model. I would like to suggest defining it.

38) Page 16, Eq. 13: What are the values for α, β, and ν?

39) Page 18, Eq. 16: The definition of orientation selectivity seems to differ from previous work where it is defined as 1 - CV = |(∑k rk ei2θk)/(∑k rk)|. I would like to suggest double checking this definition.

Supplementary material

40) Page 2: "Common for all intrinsic channels is that they conduct a single ion species." Later the authors define an intrinsic ion channel with mixed ion conductance (HCN). I suggest clarifying this point.

41) Page 4: When inputs are active, the AMPA and NMDA receptor dynamics are not clear to me. What is changing in these equations when inputs are present?

42) Supplementary Fig. 1: I found it hard to see whether the points were indeed on that plane. For clarity I would like to suggest adding 2D plots where one of the variables is fixed. For example, using the y-axis as EK+, x-axis as w, and different lines for different values of N. This might help to visualise whether a plane is capturing the points well.

43) Supplementary Fig. 3: I find it interesting that the peak firing-rate decreases as the fraction of similarly-tuned segments increase as I would expect it to be the opposite. It would be interesting to discuss these results and maybe even add to the main manuscript.

44) Page 9: Isn't the equation for ΔEion with the wrong sign? I would expect it to be after - before, which is how I understood this quantity is being used throughout the paper.

45) Page 10: "... intracellular space having a higher volume than the extracellular." Isn't this only true when considering multiple neurons in the same region (including potentially soma as well)? In other words, given a 3D region, the summed intracellular space is higher than the extracellular one, but is it also true when considering an isolated small dendritic portion?

---

## [Decision Letter · Decision Letter 2]

20 Sep 2024

Dear Nassi,

Thank you for your patience while we considered your revised manuscript "Local changes in potassium ions regulates input integration in active dendrites" for consideration as a Research Article at PLOS Biology. Your revised study has now been evaluated by the PLOS Biology editors, the Academic Editor and the original reviewers. 

In light of the reviews, which you will find at the end of this email, we are pleased to offer you the opportunity to address the remaining points from the reviewers in a revision that we anticipate should not take you very long. We will then assess your revised manuscript and your response to the reviewers' comments with our Academic Editor aiming to avoid further rounds of peer-review, although might need to consult with the reviewers, depending on the nature of the revisions.

**IMPORTANT - SUBMITTING YOUR REVISION**

*Resubmission Checklist*

*Published Peer Review*

*PLOS Data Policy*

*Blot and Gel Data Policy*

Sincerely,

Christian

Christian Schnell, PhD

Senior Editor

PLOS Biology

cschnell@plos.org

REVIEWS:

Reviewer #1: 

In this paper the authors study the consequence of localized changes in extracellular potassium concentration on dendritic excitability and neuronal responses in abstract visual cortical neuron models. In their study they did not model explicitly the changes in the extracellular K+ concentration but assumed that it is elevated specifically in some dendritic segments. The effect of the elevated [K+] was then modeled as an increase in the reversal potential for the K+ ions for the duration of the simulations (~ 1s) that influences all K+ currents including voltage sensitive and NMDA currents. 

I my previous review I raised a point regarding a fundamental assumption of the paper: that such large changes in [K+] can remain localised to a given dendritic segment. To support the claim of the authors I suggested to implement a simple simulation that combines modelling [K+] dynamics along an extended dendritic branch with realistic synaptic inputs. I argued that this simulation could reveal the potential spatial and temporal scale of the fluctuations in the [K+] concentration. I appreciate that the authors implemented such a simulation and reported the resulting changes in the K+ reversal potential. Even though I have some further comments regarding the details of these simulation, I found it already insightful: it demonstrated that within this model, changes in K+ reversal potential a localised within 30um and last for a few 100 ms. Although these simulations indicate spatio-temporally more dynamic changes in K+ reversal than that used throughout the paper, they are qualitatively similar, raising the possibility that the proposed mechanisms might actually contribute to structured fluctuations in dendritic excitability in vivo. 

Before publishing the paper I would like to ask to consider a few minor changes:

- Consider showing [K+] and not just reversal potential in the simulations in S10-S14.

- I suggest to use rate-modulated Poisson spike train (inhomogenous Poisson process) according to a slowly changing stimulus orientation. I do not see the motivation behind the choice of the Poisson distribution for the spike delay used throughout the paper - it does not correspond to neither in vivo nor in vitro stimulation of these cells. Instead, I suggest to simulate a potential in vivo spike train by first generating a trajectory of stimulus orientation, then calculating the firing rate of each neuron as a function of orientation and finally sampling spikes from a Poisson process.

- Start the simulations from an approximate steady-state of the [K+]. Now the reversal potential increases even after distributed stimulation, which makes the interpretation of the results less clear. In the case of the Poisson process stimulus the steady state (more precisely, the long term average of the [K+]) will depend on the input rates and the parameters of the K/Na pump, but can be calculated analytically or numerically.

- S10: - Is this a single simulation or the average of many simulations? If I interpret the methods correctly, each segment is divided into nodes of area 0.01 um2. Do the curves here show the average across the nodes that belong to the same segment? Can you provide information about K-dynamics within the segments? Can you add the input spikes to the graph?

- S12: increasing K_dec also affects the kinetic of the responses, but its effect (on the dynamics and the reversal potential) could depend on the steady state. This is another reason why I believe that these simulations should be started from the steady state. 

- S2 Text, Eq4; line 635, line 720: "The orientation tuning of the individual synapses was achieved by scaling the AMPA and NMDA conductance according to the synapse's tuning curve". Are these parameters really exchangeable? The strength of the synapse is a postsynaptic variable whereas its activation probability as a function of the stimulus (i.e., tuning curve) is presynaptic? The consequence of this hybrid solution is that it is not clear whether the simulations presented in the manuscript correspond to the in vivo situation (different number of spikes for synapses with preferred versus non-preferred inputs) or the in vitro situation (probably similar weights). 

- This simple model simulates diffusion in essentially 1D (along the branch - the 2nd dimension is a special because of the periodic boundary conditions). Diffusion can have qualitatively different properties in 3D, and this could be discussed.

Finally, I far as I understand the model presented in Shih et al, 2013 (Fig 2.), the changes in [K+] remain localised to within 200 nm from the synaptic cleft. This spatial scale is even smaller than the scale assumed here and would not allow cooperativity between NMDA receptors targeting the same dendritic segment. Please discuss this discrepancy.

Regarding the argument for ignoring [K+] changes caused by neighbouring segments: Vardalaki estimates that 25% of synapses are silent. Non-visually driven cells are also active, just their activity is not modulated by the visual stimulus. Cells tuned to a different orientation are not necessarily silent, they just fire with a lower rate (see for example Niell and Stryker, J Neurosci. 2008 for a quantitative spike metrics in the mouse visual cortex - I'm not aware of similar data from ferret). Furthermore, out of the 30-40 segments within 5um some could actually receive a clustered input tuned to a different orientation that would elevate the local K even if the target segment receives non-preferred input. Furthermore, the simulations with 1D diffusion (S10-14) showed that changes in [K+] can reach up to 30um. To me these considerations suggest that synapses on other segments can influence local [K], and if these changes are not correlated (e.g., state dependent changes during running versus still), then it will largely average out. This possibility should be mentioned in the discussion. 

Reviewer #2: The authors have made substantial improvements to the manuscript, and I am grateful for their thoughtful and detailed responses. I very much enjoyed reading the updated version. The complementary material added to the manuscript has strengthened the main results. I have three major comments remaining (and some minor comments that might help improve the clarity of the manuscript).

Major

1) Averaging of dendritic responses. In figs. 2, S1, and S2, the authors report averages over simulation trials, which may obscure a very interesting effect that is depicted in fig. 2g,h. In some trials, e.g., in fig. 2b, for 22.5 degrees difference, some of the grey lines show dendritic spikes while others don't. Because of that, the averages in fig. 2d might be showing intermediate values from when a spike is generated or not. Rather than plotting the average with SEM, I would like to suggest plotting the individual data points of each simulation alongside the line connecting the averages. Furthermore, SEM can also vanish for a very large number of simulations (or samples), and it would be more informative to show individual points, especially considering that the distributions may be multi-modal. It's difficult to interpret a mean with very low SEM (high confidence) if the mean does not reflect the behaviour that is most likely to occur but the intermediate behaviour between two opposite behaviours. Additionally, I think it's worth expanding the discussion about the effect of increased reliability of dendritic spikes caused by shifts in EK+, which is currently very short.

2) Equivalence between ΔEK+ changes and synaptic strength (and number of synapses). The new sentences in line 262 "This gain in the dendritic output was similar to the one expected from increasing the number of synapses (S5 Fig.)." and line 352 "This amplification was similar to the one expected by increasing the number of synapses (S8 Fig.)." reflect an important point, which is discussed in more details in the discussion section (lines 414-430). The first two sentences (lines 262 and 352), in my opinion, downplay the importance of the modulation of neuronal excitability through modulation of EK+. In other words, to get stronger weights, long-term potentiation has to occur just once (or until the high value is reached), resulting in long-lasting effects. The authors state that (line 414) "Compared to long-term plasticity mechanisms, [K+]o-dependent dendritic integration modulation requires minimum usage of resources.", but that's not the case if plasticity has to occur once to increase the weight to the optimal values. The temporal aspect of EK+ modulation is the key point here, as highlighted by the authors in the discussion section. I would like to suggest emphasising this point already in lines 262 and 352.

3) Inhibition. I appreciate the inclusion of an example of the dendritic response when inhibition is present. However, a closer look at the parameter values indicates that inhibition is very weak compared to excitation. Thus, the conclusion that (line 227) "This effect persisted in the presence of non-specific inhibitory (GABAA) input (S2 Fig.)" is too strong. To support such a claim, the authors should test the limits of the model, by either increasing the strength of the inhibitory synapses, their firing-rate, and/or number. I would expect a very strong shunting effect from strong inhibitory inputs.

Minor

4) Lines 175/176: "... increases as 1/R …" -> R is not defined. For R2=R1, ΔEK+ is undefined, as correctly pointed by the authors. However, I still find it useful to mention that it diverges as R2 approaches R1. This indicates that ΔEK+ varies from 0 to infinity.

5) Line 177: "By multiplying … regime with the …" -> Shouldn't "with" be replaced by "by"?

6) Line 187: "... a technique which averages in space and time and likely …" -> I think there is a word missing right after "averages". What is this technique averaging?

7) Line 196: "... above the baseline" -> Isn't zero the baseline for ΔEK+? If not, I would like to suggest clarifying which baseline is being referred to here.

8) Line 225: "EK+ shifts caused a linear broadening of dendritic spikes (Fig. 2c, 2d, S1 Fig.);" Fig. 2d seems to portrait a nonlinear relationship between duration and ΔEK+ (maybe quadratic?). To claim a linear relationship, the authors should add the line to the plot. Otherwise, I would suggest deleting the word "linear" from the sentence.

9) Line 253: The average synaptic activity (w) is not well defined for the point-dendrite neuron. For the statistical analysis (Fig. 1), it's clear that it normalised to the maximum response (the plot shows a.u., but isn't it normalised?). I would like to suggest avoiding arbitrary units here. Is it normalised by the maximum activity? In this case, what's the maximum activity? Referring to the methods would suffice. The methods may be improved with the inclusion of a few more details about it as well.

10) Line 259: "Using equation (3) we estimated the probability …" -> This sentence suggests that the probability of generating a dendritic spike comes directly from eq 3, but from the methods (line 681) I understood that the probability was estimated based on simulations. I would like to suggest clarifying this point.

11) Line 275: ".. I-V curve using dVm/dt=I/C" -> The negative sign is missing, as it should be -I/C for the plots in fig 3 to reflect the correct stability of the fixed points.

12) Line 276: "down-stable" is not clearly defined. I interpreted it as the stable fixed-point for the membrane potential being in the down-state (hyperpolarized), opposite to the up-stable from the up-state. It may be worth defining the term "down-stable".

13) Line 307: "... prolongs the duration of the up- and bistable states …" -> This sentence can be more precise. It prolongs the duration for which the system is described by up- and bistable states. The state of the dendrite is just one, whether it's constant, decreasing, or increasing in value depends on the input (dVm/dt=-I/C). The dendritic dynamics is described by dVm/dt which comprises fixed-point(s).

14) Line 310: Is "underlined" correct in this sentence?

15) Line 314: Additional to "boost dendritic spike generation" the authors may add the change in probability of spike generation (see point 1).

16) Line 463: Is "to or by" correct in this sentence?

17) Eq. 10: The sum symbol may not be necessary given that Iext and Iint incorporate all external and intrinsic currents.

18) Eq. 11: Vm should be subscript in the exp argument.

19) Eq. 11: The left-hand side should be Vm rather than dVm given that the update for this method is: Vm(t+dt) = V∞(t) + (Vm(t) - V∞(t))exp(-dt/τVM(t)). dVm indicates an infinitesimal difference (after - before), while eq 11 uses the solution of the differential equation when all but Vm are assumed to be constant between t and t+dt.

20) Eq. 12: What does the sum symbol indicate? The equation in the supplementary material describing synaptic currents has gsyn, B, A, and Mg(V), which seems to be missing in Eq. 12.

21) Line 691: dVm/dt = -I/C -> minus sign also missing.

22) Line 722: Is lambda the average interval between events? To the best of my knowledge, the parameter for a Poisson distribution is the frequency of events, which is usually defined as lambda. This point is just a clarification for the authors to update the text if they find it useful.

23) Fig. 1h: The y-axis labels of both plots are different. Aren't they supposed to represent the same variable?

24) Fig. 2b: The caption mentions a dotted line indicating the threshold to identify spikes. This line is not there or it's barely visible.

25) Fig. 2c: Would it be possible to include x- and y-axes to this plot with the dotted line at -30 mV to exemplify how the points in fig. 2d are calculated?

26) Fig. 2f: The separation of ΔEK+ = 0 from the rest creates a box that looks like the colour map (which is on the right of the panel), especially given that the colours change like the colour map. Would it be possible to change it to avoid confusion to the reader?

27) Fig. 3a-d: The values on the y-axis are missing.

28) Fig. 3a-d: The arrowheads are bigger in the updated manuscript, but some of them are very distorted, e.g., the blue arrowhead between -20 and 0 mV in panel d. Would there be a way to have the arrowheads without distorting them? It's difficult to see them in many parts of the plot.

29) Supplementary Figs.: In some of the plots, the units are inside square brackets, but the main figures have parentheses. Parenthesis is the standard format according to SI.

30) S1 Text: Temperature in Kelvin does not have the degree symbol.

31) S1 Text: NMDA conductance is negative, but it has to be positive if conductance is defined as the inverse of the resistance. The sign problem can be solved by removing the negative sign from the Mgblock(Vm) function.

---

## [Editor Report · Decision Letter 3]

1 Nov 2024

Dear Dr Papoutsi,

My name is Luke Smith - I am an editor at PLOS Biology and am writing on behalf of my colleague, Christian Schnell, who is off on vacation this week. Thank you for your patience while we considered your revised manuscript "Local changes in potassium ions regulates input integration in active dendrites" for publication as a Research Article at PLOS Biology. This revised version of your manuscript has been evaluated by the PLOS Biology editors and by the Academic Editor, and I am pleased to say that we are fully satisfied by the changes made in response to the previous round of review.

Based on our Academic Editor's assessment of your revision, we are likely to accept this manuscript for publication - however before we can editorially accept your manuscript we need you to address a few last data and policy related requests in another short revision. We therefore ask that you revise your manuscript in response to the following points.

EDITORIAL REQUESTS: 

1) TITLE: We think that 'regulates' should be changed to 'regulate' in your title : 

'Local changes in potassium ions regulate input integration in active dendrites'

2) DATA/CODE: Thank you for providing all of the models underlying this study on github and ModelIDB. I have a few requests/queries about this code and data. 

A - For some reason, I had trouble finding the model on ModelIDB when searching for the access number (access number: 267732) or when searching with the corresponding authors' name. Sorry if I am missing something obvious here, but can you make sure this model is accessible, and perhaps point me to it with a link? 

B - We are generally happy for you to also host your code and data on github, but, per journal policy, we cannot accept *sole* deposition of code in GitHub, as this could be changed after publication. We therefore ask that you archive this version of your publicly available GitHub code to Zenodo. Once you do this, it will generate a DOI number, which you will need to provide in the Data Availability Statement (you are welcome to also provide the GitHub access information). See the process for doing this here: https://docs.github.com/en/repositories/archiving-a-github-repository/referencing-and-citing-content

We expect to receive your revised manuscript within two weeks. 

*Published Peer Review History*

*Press*

Sincerely,

Luke

Lucas Smith, PhD

Senior Editor

lsmith@plos.org

PLOS Biology

-on behalf of-

Christian Schnell, PhD,

Senior Editor

cschnell@plos.org

PLOS Biology

---

## [Editor Report · Decision Letter 4]

13 Nov 2024

Dear Nassi,

Thank you for the submission of your revised Research Article "Local changes in potassium ions regulate input integration in active dendrites" for publication in PLOS Biology. On behalf of my colleagues and the Academic Editor, Jozsef Csicsvari, I am pleased to say that we can in principle accept your manuscript for publication, provided you address any remaining formatting and reporting issues. These will be detailed in an email you should receive within 2-3 business days from our colleagues in the journal operations team; no action is required from you until then. Please note that we will not be able to formally accept your manuscript and schedule it for publication until you have completed any requested changes.

While attending to those requests to come, please also provide the full zenodo link in the Data Availability Statement in Editorial Manager. 

PRESS

Sincerely, 

Christian

Christian Schnell, PhD

Senior Editor

PLOS Biology

cschnell@plos.org